# Unveiling the Future of Human and Machine Coding: A Survey of End-to-End Learned Image Compression

**DOI:** 10.3390/e26050357

**Published:** 2024-04-24

**Authors:** Chen-Hsiu Huang, Ja-Ling Wu

**Affiliations:** Department of Computer Science and Information Engineering, National Taiwan University, Taipei 106, Taiwan; wjl@cmlab.csie.ntu.edu.tw

**Keywords:** deep learning, image compression, video coding, neural compression, coding for machines

## Abstract

End-to-end learned image compression codecs have notably emerged in recent years. These codecs have demonstrated superiority over conventional methods, showcasing remarkable flexibility and adaptability across diverse data domains while supporting new distortion losses. Despite challenges such as computational complexity, learned image compression methods inherently align with learning-based data processing and analytic pipelines due to their well-suited internal representations. The concept of Video Coding for Machines has garnered significant attention from both academic researchers and industry practitioners. This concept reflects the growing need to integrate data compression with computer vision applications. In light of these developments, we present a comprehensive survey and review of lossy image compression methods. Additionally, we provide a concise overview of two prominent international standards, MPEG Video Coding for Machines and JPEG AI. These standards are designed to bridge the gap between data compression and computer vision, catering to practical industry use cases.

## 1. Introduction

“A picture is worth a thousand words”. Indeed, a picture conveys more meaning than text to human perception through the visual system. With the rapid development of artificial intelligence (AI) and significant advancements in computer vision, images provide critical information to machine vision beyond pure signal data. In the realm of AI, both humans and machines consume vast amounts of images and videos from various sources, including professional production, consumer creation, and IoT devices. This trend is especially pronounced in smart city scenarios with 5G networks, where surveillance cameras, drones, and self-driving cars produce videos almost continuously, surpassing user-generated content in the mobile age.

For perceiving or analyzing visual content, it is crucial to construct and represent it effectively. An image comprises a series of pixels with color components arranged in raster-scan order. The resolution and color depth of an image are vital for representing real-world objects accurately. However, higher resolution and deeper color depth lead to increased storage requirements. Images inherently contain highly redundant data, with pixels exhibiting high correlation. Therefore, effective data compression is essential for digital image applications to succeed. The types of image compression could be either lossless or lossy. Lossless image compression is often used in medical images or digital arts, where the prefect reconstruction of the original content is a must. Therefore, the compression rate is the only metric to evaluate a lossless compression method. Lossy image compression achieves more aggressive size reduction by discarding image details that are imperceptible to the human visual system. Since there will be distortions in lossy image compression, we often consider both bitrate reduction and distortion together as the rate–distortion trade-off. We evaluate a codec’s quality metrics distortions at various different bitrates and plot all the trade-offs as a *rate–distortion curve*. The JPEG standard [1] pioneered the lossy image compression techniques for its surprising compression rate and satisfying image quality in the 1990s. Following the success of JPEG, video coding standards such as H.261 [2] and MPEG-1/MPEG-2 [3] were introduced to efficiently compress video content for applications like video conferencing and storage media (e.g., VCD/DVD). Over the years, international coding standards mainly pursued high compression performance toward human consumption tasks, where minimizing signal reconstruction error is the critical metric to reflecting human perception.

In 2012, the introduction of AlexNet [4] marked a significant advancement in the object classification task on the ImageNet dataset [5], heralding the renaissance of artificial neural networks. The rapid development of deep neural networks led to numerous breakthroughs in various computer vision domains, including object detection [6] and semantic segmentation [7], effectively empowering machines with vision capabilities. However, it is noteworthy that large-scale image datasets used to train deep neural networks are typically in compressed JPEG format. By discarding high-frequency image details imperceptible to human vision, JPEG lossy compression may introduce bias in JPEG-trained machine learning models [8]. To address this bias, researchers like [8,9,10] have proposed machine learning-friendly JPEG compression frameworks. These frameworks adjust the JPEG quantization table to mitigate misclassifications caused by discarded image details. Nevertheless, even non-JPEG compressed images (e.g., HEIF [11]) can potentially trigger machine bias and lead to misbehavior. Conventional lossy codecs, initially designed for human perception, pose challenges for specialization in machine tasks.

In recent years, end-to-end learned image compression [12,13,14,15,16,17,18,19,20], or neural compression codecs have emerged and demonstrated superiority over modern conventional codecs. Unlike traditional image processing algorithms that cannot be directly applied to conventional codecs like JPEG, neural compression allows for the direct application of CNN-based methods to the compressed latent space [21,22,23,24]. This approach leverages joint compression–accuracy optimization [23] and eliminates the need for image decompression.

Research by Chamain et al. [25] illustrates the potential to significantly improve task accuracy through joint fine-tuning codec and task networks. Selective fine-tuning can be applied on the encoder, decoder, or application-dependent task networks, achieving rate–distortion optimization for human vision and rate–accuracy improvements for machine vision. In alignment with this trend, international standard committees have initiated efforts such as MPEG Video Coding for Machines (VCM) [26], and JPEG AI [27], aiming to bridge data compression and computer vision, catering to both human and machine vision needs in future explorations.

### 1.1. Prior Review Papers

Several review papers in the literature about recent developments in end-to-end learned image and video compression are summarized in Table 1. Previous reviews [28,29,30,31,32], such as that of Ma et al. [28] focus on deep learning-based image compression schemes. In contrast, Refs. [33,34,35] primarily discuss learning-based video coding techniques. Due to the high computational requirements of end-to-end learned video compression schemes, most review papers on learning-based video coding cover hybrid strategies, which integrate machine learning techniques to enhance coding tools in conventional video codecs. Notably, the introduction by Yang et al. [36] takes a comprehensive approach to data compression with neural networks, covering lossless compression techniques using generative models such as variational autoencoders and normalizing flows. Yang’s article also reviews essential background concepts in information theory (e.g., entropy coding and rate–distortion theory) for a broader machine learning audience interested in compression. Recently, Chen et al. [32] reviewed the new concept of generative compression, which focuses on realizing visually pleasing reconstructions with minimal coding costs compared with traditional codecs.

To the best of our knowledge, we are the first end-to-end (E2E) learned image compression survey paper that connects learning-based image coding with coding for visual analysis, thus unveiling the era of both coding for humans and machines. We concentrate on lossy image compression compared to prior review papers and describe how the conventional coding tools are migrated to end-to-end learned approaches in detail. We focus on lossy image compression for two main reasons: (1) most computer vision tasks operate on lossy compressed media, and (2) the richness of the media consumption experience relies on high-quality content with a high bit-rate, where lossy compression is indispensable. Prior review works primarily focus on introducing various learned compression methods without presenting the connections between traditional methods and modern learned approaches. Our work mitigates this shortcoming. Furthermore, our work also reviews visual analysis coding methods and how the visual analysis studies are evolved from feature extraction oriented to compression domain analysis and the most recent latent representation scalable coding approaches. The scalability of latent representations fulfills the coding requirements for both human-centric and machine-centric applications, which is made possible with end-to-end learned approaches.

### 1.2. Organization

This paper is organized as follows. Section 2 briefly introduces the most fundamental conventional hand-crafted image and video coding standards. Although we focus on learned image compression, we still cover conventional video codec introduction because recent image coding standards are mostly built on video intra-frame coding. Section 3 describes the end-to-end learned image compression methods developed in recent years. Since the traditional coding tools and framework developed by human intelligence provide critical guidance for the learning-based design, we discuss the migration from a conventional JPEG coding flow to the first end-to-end learned image codec in Section 3.1 and then review the subsequent improvements that advance the state of the art to surpass conventional codecs. We present the concept of joint coding for humans and machines in Section 4 and review the prior literature regarding coding for visual analysis and compressed domain analysis. In Section 4.4, we provide a concise overview of two prominent international standards, MPEG VCM and JPEG AI. Finally, we conclude our survey in the last section.

## 2. Conventional Hand-Crafted Image and Video Coding

As of today, JPEG [1] is currently the most widely used lossy image compression standard for digital images in content sharing and various image capture devices [37,38]. Since JPEG was developed in 1992, new image coding standards such as JPEG 2000 [39], WebP [40], Better Portable Graphics (BPG) [41], and High-Efficiency Image Format (HEIF) [11] have been proposed to respond to the continual expansion of multimedia applications. The WebP leverages the intra-frame coding technologies from the VP8 [42] video codec, which consists of intra-prediction, in-loop filtering, and block adaptive quantization technologies to deliver superior coding efficiency compared to JPEG. Since then, the intra-frame coding tools from a modern video codec have been reused to develop advanced still image coding standards, such as HEIF and AV1 Image File Format (AVIF). In [43], the HEIF and AVIF are reported to achieve bit-rate savings compared to JPEG, which is up to 55% and 58%, respectively.

Advanced Video Coding (AVC, or H.264) [44] is the most widely used video codec in video applications such as online streaming and video-capturing devices. Its successor, High-Efficiency Video Coding (HEVC, or H.265) [45], is becoming more famous for its nearly doubled video compression efficiency compared to H.264 [46]. The most recent international video coding standard, Versatile Video Coding (VVC), was finalized in July 2020, which improves the coding quality under the same bitrate of about 32% to 43% over HEVC [47]. Owing to the high cost and the patent licensing uncertainty of HEVC, the Alliance for Open Media (AOMedia) proposed the AOMedia Video 1 (AV1) codec, designed as an open and royalty-free video coding for video transmissions over the Internet. The AV1 has been reported to achieve superior coding gains concerning HEVC under target bitrate coding, but in general, it has increased encoding times by 20% to 30% over HEVC [48]. The VVC remains the state-of-the-art video coding standard superior to HEVC and AVC under constant quality and target bitrate coding constraints. Figure 1 shows the evolution of video coding standards in terms of compression efficiency since the 1990s.

### 2.1. Image Codec: JPEG

The success of JPEG can be attributed to its low computational complexity and its effectiveness in coding images, taking into account the human visual system. A simplified high-level JPEG coding flow is shown in Figure 2. In this process, the image is divided into 8×8 blocks, which undergo Discrete Cosine Transform (DCT) to shift the pixels from the spatial domain to the frequency domain, resulting in better image energy compaction. Then, a quantization table is used to quantize DCT coefficients, leading to reduced coefficients and sparse DCT blocks. The quantized DCT coefficients are then rearranged in zigzag order for run-length encoding (RLE), which efficiently codes the number of consecutive zeros. Finally, the DC and AC coefficients are coded separately using entropy techniques such as Huffman or arithmetic codes.

The coding tools used in JPEG include YUV color space conversion, chrominance subsampling, DCT transform coding to de-correlate pixel signals, Differential Pulse Code Modulation (DPCM) coding on DC components, scalar quantization on DCT coefficients, zigzag scanning on quantized coefficients to group zeros, run-length encoding, and finally the entropy coder. All the coding tools are hand-crafted and carefully selected, with extensive experiments considering the limited computing power required to achieve 10:1 compression performance in the 1990s. An image codec’s joint development and standardization process is effective but requires a significant amount of time and effort. With the fast development of deep neural network techniques that could end-to-end learn a complex model with an objective loss function, experts and researchers have begun to develop an end-to-end optimized image compression model that learns how to model, de-correlate, and de-redundantize the pixels directly from the training images. The design principles of JPEG, focusing on transform and predictive coding, offer a good starting point for end-to-end learned image codecs, which we will cover in Section 3.1.

### 2.2. Image Codec: HEVC Intra, BPG, and HEIF

As the successor to the widely adopted H.264 standard, HEVC significantly leaps forward in compression efficiency with advanced coding techniques. HEVC intra-coding operates by partitioning an image into smaller blocks and encoding each block independently, exploiting spatial redundancies within the image. This process involves sophisticated prediction modes, transform coding, and quantization methods to represent image content efficiently. The author of FFmpeg, Fabrice Bellard, first developed the Better Portable Graphics (BPG) [41] image codec that leverages the HEVC standard to achieve remarkable compression ratios while preserving visual fidelity. BPG represents a significant advancement over traditional formats like JPEG, offering superior compression efficiency and image quality. Later, the Moving Picture Experts Group (MPEG) developed the High-Efficiency Image File Format (HEIF) [11] as the standard container format for storing individual digital images and image sequences. HEIF can store images encoded with multiple coding formats, and the HEVC intra codec is the default image codec used with HEIF. Since 2017, Apple switched to using HEIF as the default image coding format for iPhone [49]; photos taken by iPhone are now stored as files with “*.HEIC” extension names because they contain HEVC-encoded images. Similarly, the VVC intra-coding tools can be used in HEIF format as the state-of-the-art conventional image codec. Figure 3 shows the rate–distortion curve comparison of modern image codecs since JPEG to demonstrate the coding advancements in recent years.

### 2.3. Video Codec: H.264/AVC

As image codecs such as JPEG are designed to remove spatial redundancy within a still image, the temporal redundancy in a video is a crucial characteristic that needs to be addressed for a video codec to achieve a high compression ratio. The successive video frames captured within very short time intervals have been categorized as intra-frame (I-frame) and inter-frame (P-frame or B-frame) since the introduction of MPEG-1 [3] in the 1990s. The intra-frame is self-contained and coded separately. The temporal redundancy between inter-coded frames is removed by inter-frame prediction with block-based motion estimation and motion compensation techniques. Figure 4 shows a traditional video coding flow consisting of hybrid transform-based and inter/intra-prediction coding. Instead of directly transforming the original pixels in JPEG, the residual between the source block and the motion-compensated block is transformed in video codecs, which significantly boosts the compression rate of videos. The motion vectors are entropy coded and stored in the video bitstream. It is up to the encoder’s implementation to balance the motion vector overheads and the accuracy of the inter-motion prediction.

Next to the success of MPEG-2 [50] in DVD applications, the H.264 [44], also known as Advanced Video Coding (AVC) or MPEG-4 Part 10, is currently the most widely used video codec in online streaming and video capturing. Developed by the Joint Video Team (JVT) of experts from the ITU-T Video Coding Experts Group (VCEG) and the ISO/IEC Moving Picture Experts Group (MPEG), H.264 revolutionized the field of digital video compression upon its release in 2003. H.264 employs a suite of sophisticated compression techniques, including multi-picture inter-frame prediction, variable block-size motion compensation, intra-prediction, and improved Context-Adaptive Binary Arithmetic Coding (CABAC) entropy coding, to achieve remarkable compression efficiency while preserving perceptual video quality. Unlike the DC-only prediction found in MPEG-2 Part 2, H.264 introduces intra-prediction from the edges of neighboring blocks. Intra-prediction allows for the differential coding of intra-coded pictures. The pixels of a macroblock are predicted from pixels belonging to macroblocks surrounding the current and already encoded. With the intra 4×4 mode, a macroblock is divided into 16 subblocks of 4×4 pixels, and prediction is performed independently for each subblock. Only nine prediction modes are available to simplify the optimization, which determines nine sets of parameters for the prediction function as shown in Figure 5.

### 2.4. Video Codec: HEVC

HEVC [45,51] targets to improve the coding performance by further refining its predecessor, H.264/AVC. For bettering intra-prediction, it utilizes neighboring pixels to predict the current coding block with 33 angular intra-prediction modes, the DC mode, and the planar mode. For inter-frame coding, more advanced coding tools are introduced, e.g., increasing the diversity of the PU division, most probable modes (MPMs), cutting-edge motion vector prediction (AMVP), and utilizing more interpolation filter taps for sub-sample motion compensation. Meanwhile, more advanced refinement tools for decoded frames, such as in-loop filters, deblocking filters, and sample adaptative offset (SAO), are adopted in HEVC.

### 2.5. Video Codec: VVC

Versatile Video Coding (VVC) [52] emerges as the latest milestone in the evolution of video compression standards, succeeding HEVC as the most advanced and efficient codec to date. Building upon the foundations laid by its predecessors, VVC employs a range of innovative techniques, including enhanced prediction tools, more efficient coding structures, and advanced entropy coding schemes, to achieve unprecedented compression performance. By exploiting spatial and temporal redundancies within video content, VVC significantly reduces bitrate requirements while maintaining or enhancing perceptual video quality compared to previous standards. With its adaptability to diverse video applications, ranging from ultra-high-definition video streaming to immersive virtual reality experiences, VVC paves the way for the next generation of video delivery and consumption, shaping the future landscape of multimedia communication and entertainment.

## 3. Learned Image Compression

The review paper by Ma et al. [28] discusses utilizing Multi-Layer Perceptron (MLP) networks and random neural networks for compression in the 1990s; nevertheless, they were not practically viable then. However, with the resurgence of artificial neural networks, the field of learned image compression has made significant advancements, surpassing the most recent VVC intra-coding performance in terms of PSNR and MS-SSIM [53] metrics on the selected datasets. Beginning with the work of Ballé et al. [13], who proposed the hyperprior model, there has been notable progress. This model outperforms traditional codecs like JPEG and JPEG 2000 regarding PSNR and MS-SSIM metrics. Subsequently, Minnen et al. [14] further improved coding efficiency by employing a joint autoregressive and hierarchical prior model, surpassing the performance of the HEVC codec [11]. Cheng et al. [17] introduced the use of discretized Gaussian mixture likelihoods to parameterize the distributions of latent codes and attention modules to achieve comparable performance to the latest coding standard, VVC. Guo et al. [20] proposed the causal context model, which separates the latents across channels and utilizes channel-wise relationships. Experimental results demonstrate that Guo’s compression model outperforms the VVC standard on the Kodak dataset [54] in terms of both PSNR and MS-SSIM, achieving state-of-the-art rate–distortion performance. Figure 6 illustrates the rate–distortion curves of several milestones of learned compression methods, enabling a comparison of their performance, similar to the conventional codecs in Figure 3.

The great success of learned image compression methods is promising because they offer flexibility and can be adapted to various new domains while supporting different distortion losses beyond traditional metrics like PSNR or MS-SSIM. For instance, Li et al. [55] propose NeuralCAM to compress sensor raw Bayer data (RAW) at the client side for rendering and processing, leveraging DNN-based approaches to capture underlying statistics. Wang et al. [56] propose an image compression framework optimized for subjective quality, where the distortion loss is the weighted sum of Mean Square Error (MSE) and the perceptual loss DISTS [57]. We anticipate witnessing more domain-specific data being compressed by neural compression methods without necessitating the development and standardization of a new codec standard, a process historically known to take many years. Furthermore, the internal representations of DNN-based image compression methods are amenable to modern data processing pipelines, promising fast processing and reasoning [58,59,60]. However, challenges persist in this field, which we will discuss in Section 3.8.

### 3.1. End-to-End Learning Migration

We initiate the discussion on the migration towards end-to-end learned image compression, inspired by the success and widespread adoption of Convolutional Neural Networks (CNNs) with the autoencoder architecture. Similar to the coding flow of JPEG depicted in Figure 2, a typical learned neural codec comprises an encoder–decoder pair, a quantization module, and an entropy coder. Each traditional module is sequentially replaced by a learnable neural network, leading to the construction of a learned image codec. Figure 7 illustrates a simplified autoencoder-based end-to-end learned image compression flow. Conceptually, we employ learned neural network transforms $G_{a}$ and $G_{s}$ for transform coding and train a probability model *P* to fit the distribution of transformed coefficients after quantization.

#### 3.1.1. Nonlinear Transformation

The efficacy of transform coding lies in separating the task of decorrelating a source from coding it [61]. Transform coding traditionally assumes the source to be Gaussian, as it leads to a closed-form solution. However, the transforms utilized in conventional codecs have been predominantly linear over the years due to the inherent challenges in designing nonlinear transforms. Nevertheless, the Gaussian source assumption may not always hold true, resulting in reduced coding efficiency as highlighted by Ballé et al. in [62,63].

With the advent of deep neural networks, it is well established that, given an appropriate set of parameters, DNNs can approximate arbitrary functions. Consequently, Ballé et al. [64] proposed the Generalized Divisive Normalization (GDN) transform, which comprises a linear mapping (matrix multiplication) followed by a nonlinear parametric normalization. The nonlinear transformation is defined as
(1)$$v_{i}=\frac{r_{i}}{\beta_{i}+\sum_{j}\gamma_{ij}\left|r_{j}\right|},$$
where *r* denotes the linear responses of the current layer, *v* represents the vector of normalized responses, and the vector β and matrix γ define the parameters of the transformation. The parameters of the entire transformation (linear transform, subsampling, and nonlinear parametric normalization) are optimized over a database of natural images to minimize the negative entropy of the responses. The GDN transformation is continuous and can be efficiently inverted as IGDN as described in [64]. The optimized transformation substantially Gaussianizes the data, resulting in significantly smaller mutual information between transformed components. Moreover, the transformation can be cascaded, providing an unsupervised method for optimizing a deep network architecture to convert any signal to uncorrelated latent codes. Figure 8 illustrates the use of nonlinear transforms as analysis transforms $G_{a}$ and synthesis transforms $G_{s}$, which emulate traditional image transforms.

#### 3.1.2. Factorized Entropy Model

Ballé et al. [12,13] pioneered the use of the autoencoder architecture with CNNs to end-to-end train a pair of learned forward and inverse transforms, akin to the forward DCT and inverse DCT transform in the JPEG flow. Given an input image x∈X, the neural encoder $G_{a}$ transforms *x* into a latent representation $y=G_{a}\left(x\right)$, and the neural decoder reconstructs an approximate image $\hat{x}=G_{s}\left(y\right)$. The encoder and decoder modules are denoted as the analysis transform and synthesis transform, respectively. Figure 8 illustrates the transformation network architecture proposed in [13]. While encoding the learned representation *y* directly with an entropy coder may seem straightforward, the traditional autoencoder is not optimized for compression, and the latent representation needs to be quantized to a discrete-valued vector $\hat{y}=Q\left(y\right)$ for entropy coding. The migration to an end-to-end learning approach faces several challenges: Firstly, the latent representation *y* must undergo quantization, but the quantization operation is not differentiable, making it challenging to train the network. Secondly, achieving a balanced trade-off between bit rate and reconstruction quality requires learning, but estimating the rate is not straightforward in the training optimization process.

Since the marginal probability of the discrete-valued $\hat{y}$ is unknown, learned image compression methods train an *entropy model*, which serves as a prior on $\hat{y}$ to fit the marginal probability $P_{\hat{y}}$. The simplest entropy model is a fully factorized model [12], parameterized by a set of univariate distributions $p_{y_{i}|\psi }\left(\psi \right)$: (2)$$p_{\hat{y}|\psi }\left(\hat{y}|\psi \right)=\prod_{i}\left(p_{y_{i}|\psi }\left(\psi \right)*U\left(-\frac{1}{2},\frac{1}{2}\right)\right)\left({\hat{y}}_{i}\right),$$

One approach to address the non-differentiability of quantization is with the help of Additive Uniform Noise (AUN), which involves applying uniform noise $U(-\frac{1}{2},\frac{1}{2})$ via convolution to generate $\hat{y}$ during training. During inference, integer rounding is directly applied. Various other approximation methods have been proposed, and we will delve into this aspect in Section 3.7.1.

To estimate the *rate* of the quantized discrete code $\hat{y}$, following Shannon entropy theory [65], the minimum average code length is lower-bounded by the entropy of the discrete probability $P_{\hat{y}}$: (3)$$R\left(\hat{y}\right)=H\left(P_{\hat{y}}\right)=-E\left[\log P_{\hat{y}}\right].$$

On the decoder side, we decode $\hat{y}$ from the bitstream and reconstruct the image $\hat{x}=G_{d}\left(\hat{y}\right)$ using the neural decoder. The *distortion* $D(x,\hat{x})$ is measured by a distortion metric $d(x,\hat{x})$, where MSE and MS-SSIM are commonly used. We optimize the network parameters for a weighted sum of the rate and distortion terms using a Lagrange multiplier λ to control the rate–distortion trade-off: (4)$$L=R+\lambda D=-E\left[\log P_{\hat{y}}\right]+\lambda d\left(x,\hat{x}\right).$$

Unlike JPEG, which does not inherently perform rate–distortion optimization to minimize distortion within a bit budget constraint *B*: (5)$$\min_{R<B}D\left(x,\hat{x}\right).$$

End-to-end optimized rate–distortion trade-off effectively balances bitrate and reconstruction quality at various λ values. Consequently, we can specify different rate–distortion trade-offs for applications targeting various bit-rate scenarios. The concept of the factorized model is illustrated in Figure 9.

Until now, the challenges outlined previously have been addressed in [12], enabling the end-to-end training of learned image encoders/decoders over any given dataset without the need for cherry-picking coding tools or hand-crafted designs. Subsequently, numerous learning-based compression methods have been proposed and have progressed to state-of-the-art performance compared to conventional image codecs [13,14,17,18,19,20]. These works, in general, follow a similar scheme but may vary in their entropy model and transform network structure.

#### 3.1.3. Hyperprior Entropy Model

While the factorized entropy model successfully fits the marginal distribution of $P_{\hat{y}}$ with a fully factorized model, it has been observed in [13] that strong spatial dependencies exist among the elements in the latent representation *y*. To further reduce spatial redundancy in the latent code and predict its marginal distribution more accurately, Ballé et al. [13] proposed the hyperprior entropy model, illustrated in Figure 10. The structural information in the compressed latent *y* is further analyzed with an analysis transform $h_{a}$ (hyper encoder) to generate side-information $z=h_{a}\left(y\right)$, which is then quantized to $\hat{z}=Q\left(z\right)$ and entropy coded along with $\hat{y}$ into the bitstream.

The hyperprior model $p_{\hat{y}|\hat{z}}\left(\hat{y}|\hat{z}\right)$ represents the estimated distribution of $\hat{y}$ conditioned on $\hat{z}$. To model $\hat{z}$ for conducting entropy coding, we employ a fully factorized model similar to Equation (Equation 2). Further, we model the distribution $p_{\hat{y}|\hat{z}}\left(\hat{y}|\hat{z}\right)$ as a Gaussian distribution with mean μ and scale σ provided by a trained synthesis transform $h_{s}$ (hyper decoder): (6)$$\begin{array}{cc} p_{\hat{y}|\hat{z}}\left(\hat{y}|\hat{z}\right) & =\prod_{i}p_{\hat{y}|\hat{z}}\left({\hat{y}}_{i}|\hat{z}\right)∼N\left(\mu ,\sigma^{2}\right) \end{array}$$(7)$$\begin{array}{cc} \mu ,\sigma & =h_{s}\left(\hat{z}\right) \end{array}$$

The hyperprior entropy model, incorporating Generalized Divisive Normalization (GDN), marks the first end-to-end learned compression method that outperformed traditional codecs like JPEG and JPEG 2000 in terms of PSNR and MS-SSIM metrics. Although its coding efficiency lags behind modern image codecs such as HEVC, the hyperprior model remains a promising learned image compression model due to its relatively simple model design and less computationally intensive requirements.

### 3.2. Context Models

#### 3.2.1. Pixel Probability Models

As discussed in Section 2, traditional image and video codecs use both transform and predictive coding techniques. The autoencoder architecture methods introduced in Section 3.1 primarily focus on transform coding. In general, predictive coding utilizes historical data to predict future data. It encodes only the residual, which is the difference between the predicted value and the actual value, thereby reducing the magnitude of the data. For instance, JPEG encodes the difference between neighboring DC coefficients, $z_{i,j}^{DC}-z_{i,j-1}^{DC}$, in a raster-scan order, implementing a first-order Markov model.

In learned compression, the *context model* employs machine learning techniques to model the statistical dependencies between symbols in the data. The pixel probability model, a pixel-based context model, operates in the spatial domain as illustrated in Figure 11. Methods such as PixelCNN [66] and PixelRNN [67] model the joint distribution of pixels over an image x using the following product of conditional distributions, where $x_{i}$ represents a single pixel:(8)$$p\left(x\right)=\prod_{i}^{N^{2}}p\left(x_{i}|x_{1},...,x_{i-1}\right).$$

Due to their autoregressive nature, pixel probability models encode one pixel at a time conditioned on previously generated pixels. This sequential generation process is slow for high-resolution images and cannot be easily parallelized. Moreover, in addition to the high computational complexity, these models need help in capturing long-range dependencies and global structures in images, often resulting in artifacts and suboptimal coding efficiency. Consequently, pixel probability models were proposed in the early stages of learned compression development and needed to achieve promising coding efficiency for lossy image compression. Readers interested in further details may refer to [68,69,70,71].

#### 3.2.2. Joint Context and Entropy Model

The context model can be applied to the transformed latent representation. Minnen et al. [14] enhance the hyperprior model by incorporating the context, meaning they utilize both $\hat{z}$ and the context to predict the probability of each entry of $\hat{y}$. More specifically, a more accurate entropy model can be developed by predicting μ and σ conditioned on both $\hat{z}$ and the causal context of previously decoded ${\hat{y}}_{<i}$. This fact can be expressed as:(9)$$\begin{array}{cc} \psi & =h_{s}\left(\hat{z}\right) \end{array}$$(10)$$\begin{array}{cc} \phi_{i} & =f_{cm}\left({\hat{y}}_{<i}\right) \end{array}$$(11)$$\begin{array}{cc} \mu_{i},\sigma_{i} & =f_{ep}\left(\psi ,\phi_{i}\right) \end{array}$$(12)$$\begin{array}{cc} p_{\hat{y}|\hat{z}}\left({\hat{y}}_{i}|\hat{z}\right) & =\prod_{i}\left(N\left(\mu_{i},\sigma_{i}^{2}\right)*U\left(-\frac{1}{2},\frac{1}{2}\right)\right)\left({\hat{y}}_{i}\right), \end{array}$$
where $h_{s}$ is the hyper-decoder, $f_{cm}$ denotes the context model, and $f_{ep}$ represents the entropy model. In practice, the context model utilizes a limited context with a 5×5 convolution kernel to reduce computational complexity. The joint context and hyperprior model can be illustrated in Figure 12. Minnen’s approach reportedly outperforms the HEVC codec.

Since the hyperprior entropy model can more accurately estimate the distribution of latent codes with a generalized context model form, many works have been proposed to improve context models. Cheng et al. [17] suggested employing discretized Gaussian mixture likelihoods to parameterize latent codes’ distribution and leverage attention modules to enhance performance. In comparison to [13,14], the Gaussian mixture model offers more flexibility in achieving arbitrary likelihoods:(13)$$p_{\hat{y}|\hat{z}}\left({\hat{y}}_{i}|\hat{z}\right)=\left(\sum_{k}w_{i}^{\left(k\right)}N\left(\mu_{i}^{\left(k\right)},\sigma_{i}^{\left(k\right)}\right)*U\left(-\frac{1}{2},\frac{1}{2}\right)\right)\left({\hat{y}}_{i}\right)$$
where *k* represents the index of mixtures. Each one of the mixtures is characterized by a Gaussian distribution with parameters including weights $w_{i}^{\left(k\right)}$, means $\mu_{i}^{\left(k\right)}$, and variances $\sigma_{i}^{\left(k\right)}$ for each latent code ${\hat{y}}_{i}$. Additionally, the attention module is incorporated to ensure that the learned models focus more on complex regions, allocating more bits to challenging parts and reducing the bits allocated to simpler ones. Cheng’s model reportedly achieves comparable performance with the latest compression standard VVC in terms of PSNR.

Lee et al. [15] introduced the context-adaptive entropy model, which conditions both bit-consuming contexts $c^{\prime }$ and bit-free contexts $c^{\prime \prime }$. Rather than estimating the mean and scale through the hyper-decoder $h_{s}$ in Equation (7), Lee’s approach revises $h_{s}$ to generate the context $c^{\prime }$ as side-information coded in the bitstream:(14)$$\begin{array}{cc} \mu_{i},\sigma_{i} & =f_{ca}\left(c_{i}^{\prime },c_{i}^{\prime \prime }\right) \end{array}$$(15)$$\begin{array}{cc} c_{i}^{\prime } & =E^{\prime }\left(c^{\prime },i\right),\;c^{\prime }=h_{s}\left(\hat{z}\right) \end{array}$$(16)$$\begin{array}{cc} c_{i}^{\prime \prime } & =E^{\prime \prime }\left(\langle \hat{y}\rangle ,i\right), \end{array}$$
where $f_{ca}$ represents the context-adaptive entropy model, $\langle \hat{y}\rangle$ denotes the known and entropy-coded subset of $\hat{y}$, $E^{\prime }$ extracts $c_{i}^{\prime }$ from $c^{\prime }$, and $E^{\prime \prime }$ extracts $c_{i}^{\prime \prime }$ from received $\langle \hat{y}\rangle$. Lee’s model is context-adaptive because it relies on prior-coded bitstream, and the context $c^{\prime \prime }$ is essentially bit-free.

Mentzer et al. [72] opted to model the entropy of the latent representation by employing a 3D-CNN context model, which learns a conditional probability model of the latent distribution of the autoencoder. The context model $P(\hat{z})$ estimates each term $p(\hat{z_{i}}|{\hat{z}}_{i}-1,\ldots ,{\hat{z}}_{1})$:(17)$$P_{i,l}\approx p\left({\hat{z}}_{i}=c_{l}|{\hat{z}}_{i}-1,\ldots ,{\hat{z}}_{1}\right),$$
where $P_{i,l}$ specifies the probabilities of each latent code with l=1,…,L for every 3D location *i* in $\hat{z}$. Ma et al. [73] extended this concept to the cross-channel context, which utilizes cross-channel correlations. Their approach divides the latents into several groups according to the channel index. It codes the groups individually, where previously coded groups are utilized to provide cross-channel context for the current group. The experiments demonstrate that the cross-channel model achieves BD-rate reductions of 2.5% over the VVC codec on the Kodak dataset. Figure 13 illustrates the different context model comparisons mentioned above.

Minnen and Singh [74] proposed the channel-wise autoregressive entropy model to address the slow decoding time issue of spatial autoregressive models [14,72]. The channel-conditional model splits the latent tensor along the channel dimension into *N* roughly equal-sized slices and conditions the entropy parameters for each slice on previously decoded slices. Experiments indicate that more slice splits in the latent tensor allow more opportunities to model the dependencies between channels and reduce entropy. However, this comes at the cost of additional computation.

Similarly, Guo et al. [20] introduced the causal context model, which separates the latents across channels and utilizes channel-wise relationships to generate highly informative adjacent contexts. They developed the causal global prediction model to find global reference points for accurate predictions of non-decoded points akin to infra-frame block pattern matching. Specifically, a correlation matrix is calculated from the first channel group to describe global dependencies. This causal correlation matrix guides the global prediction for these non-decoded channels. Experimental results demonstrate that the causal context compression model outperforms the standard VVC codec on the Kodak dataset regarding both PSNR and MS-SSIM.

### 3.3. Multi-Scale Approaches

As JPEG 2000 [39] employs wavelet transform to capture a multi-resolution representation of the image, several learning approaches have adopted similar concepts for multi-scale image analysis and coding. Rippel et al. [75] initially introduced a pyramidal decomposition architecture resembling wavelets, followed by an inter-scale alignment network. Their method utilizes bit plane coding and adaptive arithmetic coding, providing a lightweight solution capable of real-time operation. Additionally, they incorporate a discriminator loss, in addition to the reconstruction loss, to achieve visually pleasing results.

Hu et al. [18] propose a coarse-to-fine framework with hierarchical layers of hyper-priors, serving as a computationally efficient alternative to context models such as [14,15]. They design two layers of the Signal Preserving Hyper Transform cascaded after the analysis transform of [13] to facilitate multi-layer structure analysis. Subsequently, the layer-1 and layer-2 hyper-information are combined with the synthesis transform output in the Information Aggregation Reconstruction sub-network, leveraging side information to enhance quality. In Hu’s survey and benchmark report [29], the coarse-to-fine model achieves competitive rate–distortion performance against VVC and context-model-based methods while experiencing a significant acceleration in encoding and decoding time.

Ma et al. [16] proposed the iWave++ scheme, comprising a trained wavelet-like transform that converts images into coefficients without any information loss. Unlike prior works, focusing solely on lossy compression and requiring one model for each rate–distortion trade-off, iWave++ supports both lossless and lossy compression in one model and achieves arbitrary compression ratios by adjusting the quantization scale. Experimental results demonstrate that iWave++ achieves state-of-the-art compression performance for both lossy and lossless compression despite its significantly higher computational complexity compared to [15]. However, a fast version is provided with a lightweight context model and block parallel coding strategy. In summary, the aforementioned multi-scale approaches differ substantially from the joint context and hyperprior model described in Section 3.2.2, yet they demonstrate competitive coding efficiency with less computational complexity.

Recently, Duan et al. [76] addressed the problem of lossy image compression from the perspective of generative modeling, starting with ResNet Variational AutoEncoders (VAEs). They redesigned the probabilistic model as QResNet VAEs, utilizing uniform posteriors and a Gaussian distribution convolved with a uniform distribution for modeling the prior to enable quantization-aware training. The hierarchical VAE model compresses images in a coarse-to-fine fashion and supports parallel encoding and decoding. Experiments reveal that hierarchical VAEs can achieve state-of-the-art compression performance with low computational complexity.

### 3.4. Attention Module and Content-Based Approaches

The human visual system is known to distribute attention unevenly across different parts of an image, resulting in spatially variant local content information. Li et al. [77] were the first to utilize a content-weighted importance network to generate an importance map, guiding the allocation of local bit rates. Through joint rate–distortion optimization, the quantized attention mask automatically balances between entropy and distortion, with the network learning to output more predictable (low entropy) symbols for smooth regions while allowing for high entropy symbols in more complex regions. Mentzer et al. [72] expanded the encoder output instead of using a separate network for their importance mask, applying pointwise multiplication with the latent variable z. Additionally, Chen et al. [19] applied multi-scale attention masks at different layers atop their 3D-CNN context model to adaptively weigh latent features for compression. Specifically, they proposed the Non-Local Attention Module, consisting of ResBlocks for attention mask generation. Chen’s ablation study shows that the successive removal of the mask branches gradually leads to a corresponding drop in PSNR quality.

In contrast to masking latent representations with attention maps, Cheng et al. [17] proposed a simplified attention module, and they inserted it within the analysis and synthesis transform of their Gaussian mixture context model. This alternative approach aims to guide the transform network to focus more on complex regions and reduce bits in simple regions. Generally speaking, adopting attention modules are a straightforward strategy to enhance rate–distortion performance. However, the gains may not be significant if the context and entropy model are already complex enough to model the data accurately.

### 3.5. Transformer Based Coding

Most existing learned image compression models are based on CNNs. Despite their great success, works focusing on more expressive transforms result in a better rate–distortion trade-off. Inspired by recent progress in Vision Transformer (ViT) [78] and Swin Transformer [79], transformer-based image compression schemes have become a new branch of learned transform research [80,81,82,83]. Zhu et al. [80] demonstrated that nonlinear transforms built on Swin-transformers can achieve better compression efficiency than CNN-based transforms, requiring fewer parameters and shorter decoding time. The SwinT-ChARM model they proposed outperforms VVC at comparable decoding speed, marking a significant milestone in learning-based methods. Liu et al. [82] proposed a transformer-CNN mixture architecture with controllable complexity to incorporate the local modeling ability of CNNs and the non-local modeling ability of transformers.

Motivated by the recent success of prompt tuning techniques [84,85], Kao et al. [83] proposed a transformer-based learned image codec that is able to trade off one image quality objective for another (e.g., PSNR versus LPIPS [86]) using a single, shared model. They introduced prompt tokens to condition the transformer-based autoencoder, generated adaptively based on the user’s preference and input image through learning a prompt generation network.

### 3.6. Generative Adversarial Network Based Coding

As human eyes are less sensitive to pixel-wise distortion in regions of complex texture, Generative Adversarial Networks (GANs) [87] can be employed to synthesize such areas effectively with low-bitrate representations. Santurkar et al. [88] first proposed the concept of *generative compression*, which uses a DCGAN-style [89] generative model to compress data. Figure 14 shows their proposed architecture.

First, a generator network (decoder) $G_{\phi }:Z\to X$ is pre-trained using an adversarial loss concerning the discriminator network, $d_{\vartheta }:X\to \left\{0,1\right\}$. Second, an encoder network $f_{\theta }:X\to Z$ is trained to minimize the distortion loss $L_{d}\left(x,\hat{x}\right)$, where $\hat{x}=G_{\phi }\left(f_{\theta }\left(x\right)\right)$ is the reconstructed image. To improve the plausibility of the reconstruction, the distortion loss function is the weighted sum of pixel-level difference and perceptual loss metric derived from deep convolution network features:(18)$$L_{d}\left(x,\hat{x}\right)=\lambda_{1}||x-\hat{x}{\left|\right|}_{2}+\lambda_{2}\left|\right|{\mathrm{conv}}_{4}\left(x\right)-{\mathrm{conv}}_{4}\left(\hat{x}\right){\left|\right|}_{2},$$
where ${\mathrm{conv}}_{4}$ is the fourth convolutional layer of an ImageNet-pretrained AlexNet [4]. The bitrate reduction is achieved by either reducing the length of the latent vector or reducing the number of bits to represent each element. Santurkar’s generative model can compress images up to a 1000X bitrate reduction but still present quality results that are semantically meaningful to humans. However, in their demo cases on the CelebA [90] dataset, those decoded faces are, in fact, not the original person anymore, though these faces look visually pleasing.

Rippel et al. [75] also proposed utilizing the discriminator loss function in an end-to-end framework to improve visual quality. They proposed a multi-scale adversarial training model to aggregate information across different scales. In contrast to Santurkar’s generative model, which is limited to small image datasets, the extreme learned image compression introduced by Agustsson et al. [91] is a GAN-based framework for full-resolution image compression. Agustsson’s extreme compression system targets bitrates below 0.1 bpp and provides two operation modes: generative compression and selective generative compression. The selective generative compression requires a semantic label map; thus, it can fully synthesize unimportant regions such as streets and trees from the label map, further reducing the bitrate significantly.

Tschannen et al. [92] propose optimizing the rate–distortion trade-off under the constraint that the reconstructed samples follow the training data’s distribution. Starting at zero bitrates, the Distribution-Preserving Lossy Compression (DPLC) learns a generative model of the data; then, at high enough bitrates, it achieves perfect reconstruction. It smoothly interpolates intermediate bitrates between learning a generative model and perfectly reconstructing the training samples. Mentzer et al. [93] further proposed the high-fidelity generative image compression model (HiFiC), which can operate in a broad range of bitrates and be applied to high-resolution images. In general, if a discriminator loss or a perceptual loss is used in the end-to-end optimization, the reconstructed image usually cannot be evaluated by traditional PSNR because the per-pixel distortion will be significantly larger than conventional codecs. The structural quality metric, such as MS-SSIM or subjective studies, is often used to evaluate the GAN-based approaches. However, the need for a universal objective quality metric remains an open problem when comparing various GAN-based image compressions fairly.

Another issue with GAN-based image compression is that the semantic regions, such as small human faces or texts, may be seriously distorted as shown in Figure 15. The discriminator cannot tell the difference between real face and distorted faces, but human vision is very sensitive to faces. Although GAN-based approaches usually deliver surprisingly visually pleasing results, they are still limited in specific applications where the semantic structures are unimportant. Especially in the coding for machine paradigms, those semantic regions are equally crucial to machine vision and cannot be compromised.

In the last two years, the breakthrough development of generative models, such as Contrastive Language-Image Pre-Training (CLIP) [94] and diffusion model [95], has given rise to the explosion of Artificial Intelligence Generated Content (AIGC). A new concept of near-realism image compression [96,97] or perceptual compression [98] is being introduced, aiming to compress the image with ultra-low bitrate while maintaining semantically consistent and visually pleasing results. Jiang et al. [96] adopted the image–text attention module to introduce text information into the codec as prior information. The text description can help the codec achieve a compact feature representation and better feature enhancement even at extremely low bit rates. Careil’s [98] perceptual compression combines a vector quantized VAE-liked encoder and a diffusion-based decoder, conditioning on textual image descriptions to generate realistic reconstructions at low bitrates. Although the perceptual compression model generates high-quality and visually pleasing images, the objects within the context are, in fact, not the original ones in the source image.

### 3.7. Miscellaneous Topics

Except for designing a more accurate and flexible entropy model that can better parameterize the distributions of the latent features, Lee et al. [99] propose a novel joint learning scheme of compression and quality enhancement Grouped Residual Dense Networks (GRDNs) [100]. Their proposed JointIQ-Net combines an image compression sub-network and a quality enhancement sub-network in a cascade; both are end-to-end trained in a combined manner.

Several research studies [101,102,103] have focused on developing better transformations between the image and latent feature space. Autoencoder-based frameworks often neglect some information during encoding, leading to unrecoverable loss during decoding. One possible solution to address this abovementioned issue is integrating Invertible Neural Networks (INNs) [104] into the encoding–decoding process, as INNs possess the strictly invertible property. Xie et al. [103] proposed an enhanced Invertible Encoding Network with INNs to mitigate information loss for improved compression. Experimental results demonstrate that the enhanced INN approach outperforms state-of-the-art learned image compression methods and VVC compression standards, especially on high-resolution image datasets. Furthermore, Ho et al. [105,106] attempted to use Augmented Normalizing Flows (ANFs) [107] for lossy image compression, termed ANFIC. ANF is a specific type of normalizing flow model that augments the input with independent noise, enabling a smoother transformation from the augmented input to the latent space.

In the learned image compression field, various models have been utilized as the entropy model, including autoregressive, softmax, logistic mixture, Gaussian mixture, and Laplacian models. Fu et al. [108] argued that using a single model for all images is suboptimal and proposed a more flexible discretized Gaussian–Laplacian–logistic mixture model (GLLMM). The GLLMM model adapts to different contents in different images and regions of one image more accurately and efficiently. Their experiments demonstrate state-of-the-art coding performance, although the encoding and decoding times remain high due to using the autoregressive context model.

#### 3.7.1. Quantization

As the quantization operation is one of the lossy components in learned image compression, correctly approximating quantization in the end-to-end optimization process remains an interesting topic. Theis et al. [109] first proposed the Straight-Through Estimator (STE), which conducts normal rounding in the forward pass but passes gradients as identity in the backward pass. Ballé et al. [12] proposed replacing quantization with Additive Uniform Noise (AUN),
(19)$$Q\left(y\right)=y+U\left(-\frac{1}{2},\frac{1}{2}\right).$$

Agustsson and Theis [110] introduced the Soft Quantization (SQ), defined as:(20)$$\begin{array}{cc} s_{\alpha }\left(y\right) & =\left\lfloor y\right\rfloor +\frac{1}{2}\frac{\tanh(\alpha r)}{\tanh(\alpha /2)}+\frac{1}{2} \end{array}$$(21)$$\begin{array}{cc} r & =y-\left\lfloor y\right\rfloor -\frac{1}{2} \end{array}$$
where ⌊·⌋ denotes the floor function, and α is a constant parameter.

Several works have explored hybrid quantization strategies in their compression schemes. For instance, Lee et al. [15] proposed utilizing AUN for entropy models and STE for synthesis transforms during training. Guo et al. [111] introduced a novel soft-then-hard quantization strategy, which first learns an expressive latent space softly and then freezes the encoder, closing the train–test mismatch with hard quantization.

Alternatively, Cai et al. [112] presented an iterative non-uniform quantization scheme, wherein the quantizer and encoder–decoder are alternately optimized. When the encoder–decoder is fixed, a non-uniform quantizer is optimized based on the distribution of representation features. Subsequently, the encoder–decoder network is then updated by fixing the quantizer.

Tsubota et al. [113] conducted a comprehensive study on existing approximations for uniform quantization, exploring different combinations. Through experiments, they concluded that approximations by adding noise are better than rounding. They further recommended a rounding-based approximation for the distortion term (decoder) and a uniform-noise-based approximation for the rate term (entropy model).

#### 3.7.2. Variable Rate Model

As we described the end-to-end learning migration in Section 3.1.2, rate–distortion optimization (RDO) is a critical component in lossy image compression, expressed as R+λD. Most learning-based approaches must train separate models with different λ values to attain optimal rate–distortion performance for matching target bitrates. However, the requirement of multiple models across low, medium, and high bitrates proves impractical for real-world deployment. Consequently, works have emerged to support multiple quality levels with a single model.

Inspired by traditional scalable video coding frameworks, scalable learned compression schemes [114,115,116] have been proposed, generating varying quality levels based on the layers of received bitstreams. Jia et al. [114] introduced a scalable autoencoder (SAE) image compression network to mitigate the necessity of training multiple models for different bitrate points. The SAE-based deep image codec comprises hierarchical coding layers as the base and the enhancement layers. Mei et al. [115] proposed a quality and spatial scalable image compression (QSSIC) model in a multi-layer structure, where each layer generates one bitstream corresponding to a specified resolution and image fidelity. This scalability is achieved by exploring the potential of feature-domain representation prediction and reuse. Lu et al. [116] presented a progressive neural image compression scheme with nested quantization. The nested quantization defines multiple quantization levels with nested quantization grids, progressively refining all latents from the coarsest to the finest quantization level.

Choi et al. [117] proposed a variable-rate image compression network with a conditional autoencoder. Two rate control parameters, the Lagrange multiplier and the quantization bin size, are the network’s conditioning variables. Chen et al. [118] proposed embedding a quality scaling factor $\mathrm{SF}\in \{a_{i},b_{i}\}$ in a learned compression network to scale latent codes. Sun et al. [119] introduced an Interpolation Variable-Rate (IVR) network by incorporating an Interpolation Channel Attention (InterpCA) module in the compression network. Two hyperparameters for rate control and linear interpolation are used to fine rate intervals in the IVR network.

Shi et al. [120] proposed a variable-rate compression scheme with a shallow to deep hyperprior network trained with a uniform quality map. The trade-off parameter λ of the rate–distortion optimization generates the quality map. Inspired by the fact that a uniform quality map can continuously scale the latent representations with a continuous λ, they constructed an individual *R*-λ model characterizing the relation between *R* and λ.

### 3.8. Open Challenges

In recent years, learning-based lossy image compression schemes have indisputably surpassed the performance of hand-crafted image codecs, though each method has pros and cons. Before discussing the challenges in this field, we summarized some of the previously mentioned techniques by rough categories in Table 2. Note that some methods are not listed simply because we have difficulty categorizing them or because they represent some specific improvements that may not generalize to others. Furthermore, for some categories without cons listed, it does not mean that they are the perfect direction to proceed. This indicates that more effort should be invested in discovering more possibilities.

Presently, the field faces two significant challenges [121,122]: computational complexity and subjective image quality. The neural compressor employs high-capacity networks to model data dependencies end-to-end, leading to improved bitrate–distortion (BD) efficiency. However, methods such as the channel-conditional approach proposed by Minnen et al. [74] achieve performance close to VVC but come at the cost of high computational complexity (600K FLOPS/pixel), making them almost three times slower on GPU compared to VTM running on CPU.

Regarding image quality, Valenzise et al. [123] conducted subjective tests on DNN-based methods and observed that these methods produce artifacts that are challenging to evaluate using traditional metrics like PSNR. They concluded that PSNR is inadequate for assessing DNN-based methods. Developing a perceptual metric closely aligned with human subjective rating remains an unresolved topic. Research efforts have been devoted to addressing the two challenges, and we will introduce some of them in this section.

#### 3.8.1. Computational Complexity

To reduce the complexity of end-to-end learned image compression, earlier work by Johnston et al. [124] applied automatic network optimization techniques to reduce the computational complexity of the hyperprior network. They used an implementation of MorphNet [125] to find the best architectures for the hyperprior network, reducing decoder run-time requirements by over 50%, thereby closing the gap towards real-time decoding since Rippel’s [75] multi-scale compression scheme.

Another complexity issue stems from the autoregressive nature of joint context and entropy models [17,20,72,73], which are computationally complex, challenging to parallelize and run sequentially on CPUs during decoding. Context models such as [20,73] surpass the VVC codec at that cost of high complexity, and generally take 20–30 s to decode a Kodak image. To improve parallelism, He et al. [126] proposed checkerboard convolution as a parallel replacement for the serial autoregressive context model. Minnen et al. [74] suggested using channel-wise context models instead of serial-decoded spatial contexts as proposed in [14]. From Figure 16, we can see that the reduced decoding time shows an order of magnitude speedup.

He et al. [127] proposed ELIC with an uneven grouping scheme to accelerate the channel-conditional method. By combining space-channel context models with residual transform networks, the ELIC model achieves a better balance between coding efficiency and execution time, attaining state-of-the-art performance. The encoding and decoding time of ELIC are nearly symmetric, taking less than 100 ms on GPU. More recently, Wang et al. [128] proposed an Efficient single-model Variable-bit-rate Codec (EVC), which achieves a decoding rate of 30 FPS with 768 × 512 input images while still outperforming VVC. They trained small neural compression models with a teacher model to reduce the model’s capacity. Specifically, they introduced mask decay to transform the large model’s parameters into a small model. Mask layers are first inserted into a pre-trained teacher model and then sparsified until all layers become the student’s structure. A miniature model capable of 30 FPS decoding for 1080P input images is obtained by reducing the complexities of both the encoder and the decoder.

We compare the speed of some key image compression methods by combining the tests conducted by He et al. [127] in Figure 16. In Figure 16, the y-axis represents the BD-rate savings over VVC, where a negative value indicates superior coding efficiency compared to VVC. The x-axis denotes the decoding time on a logarithmic scale. At present, both EVC and ELIC demonstrate superior coding performance compared to VVC, with decoding speeds approaching VVC. However, it is worth noting that the evaluation of the learned methods utilizes GPU, whereas VVC decoding is performed on the CPU.

Recently, Yang et al. [129] attempted to address the performance issue with asymmetric encoding and decoding complexity by adopting shallow or linear decoding transforms. Through empirical study [130], comparing nonlinear synthesis transform and traditional transform coding, they found that JPEG-like synthesis can perform similarly to a deep linear CNN. The JPEG-like shallow decoder is then equipped with powerful encoder networks and iterative encoding, achieving rate–distortion performance competitive with the established mean-scale hyperprior [13] at less than 50K decoding FLOPs/pixel, reducing the baseline’s overall decoding complexity by 80%. Yang’s work provides new insights into exploiting the asymmetrical nature of encoding and decoding by pairing a lightweight decoder with a powerful encoder, jointly trained to obtain high rate–distortion performance while enjoying low decoding complexity.

#### 3.8.2. Subjective Image Quality

Under low bitrate, the autoencoder-based compression methods produce blurry details due to rate–distortion optimization. Although generative compressions are visually realistic, image details tend to be distorted and unsuitable for semantic regions like faces and texts. Both types of distortion have pros and cons. Therefore, compression methods have been proposed to simultaneously optimize rate, mean-square loss, and VGG loss [131] (an adversarial loss), which can be considered a rate–distortion–perception trade-off. Kirmemis et al. [132] proposed a practical approach to perform perception–distortion analysis at a fixed bitrate, then determine the best perception–distortion trade-off point at that fixed rate.

Agustsson et al. [133] tried to mitigate this dilemma from another perspective. They proposed multi-realism compression to allow the receiver to choose a realism preference β when decoding the reconstruction $\hat{x}$ from one single latent representation $\hat{y}$. The preference controls either a low mean squared error reconstruction, a realistic reconstruction with high perceptual quality, or mixed quality in between. Unlike conventional codecs, this distortion–realism trade-off allows the learned compression to be widely adopted in various application scenarios.

Given that a universal objective quality metric remains undiscovered, we see a surge in research that considers the realism of the reconstructed image for better subjective quality [134,135,136,137].

## 4. Coding for Human and Machine Vision

We view a machine learning task as extracting the most critical features relevant to a specific task from raw data, creating a latent representation of the original data. Then, a downstream module, such as a classifier or a decoder, operates on this latent representation for decision-making (e.g., object classification) or restoration (e.g., image denoising). The best model for a machine learning task analyzes and extracts only the relevant features, analogous to the Minimum Description Length (MDL) principle [138] in information theory. In MDL, the model and the data are treated as a *code*, and the objective is to minimize the total code length required to describe both the model and the data. Finding the best model under MDL involves finding the model that most effectively compresses the data. Hence, commonalities exist between computer vision and compression tasks, offering significant opportunities for combined visual analysis and compression.

Chamain et al. [25] validated this opportunity by a studying an end-to-end framework for efficient image compression and remote machine task analysis. They demonstrated that it is possible to significantly improve the task accuracy by jointly fine-tuning the codec and the task networks, particularly at low bitrates. Their study concluded that selective fine-tuning can be applied only to the encoder, decoder, or task network; however, rate–accuracy improvements can still be achieved over an off-the-shelf codec and task network. Chamain’s research underscores a clear advantage of learning-based codecs: the optimized compression for machines.

### 4.1. Coding for Visual Analysis

The relationship between visual analysis (machine vision) and compression (human vision) has been extensively studied prior to the emergence of learning-based image coding approaches [139,140,141]. Redondi et al. [139] compared two paradigms of visual analysis: Compress-then-Analyze (CTA) and Analyze-then-Compress (ATC), finding that the best approach depends on the network conditions. They observed that the ATC paradigm yields good results at low bitrates, whereas the CTA paradigm achieves the best performance. Considering a visual retrieval application, Zhang et al. [140] proposed a joint compression scheme for feature descriptors and visual content, representing a Hybrid Analyze-then-Compress (HATC) approach. We depict the compression and analysis frameworks for visual retrieval applications in Figure 17, which consist of two-step approaches in the early days. Duan et al. [141] introduced a video coding for machines framework by incorporating coding of multiple streams, including videos, features, and models, aiming to address the multi-tasks of human perception and machine intelligence. These previous works realized machine vision on a separate feature bitstream before compressing it.

Zhang et al. [142] first posed the following question: Can the bitstream of images and image features be unified and served for compression and retrieval simultaneously? They proposed a content-based image retrieval system where images are encoded only once, and the coded bitstream can be used not only for image reconstruction but also for direct comparison to search for similar images. Several studies [8,9,10] investigated the design of bitrate-efficient quantization for image compression with minimal classification accuracy loss. These joint rate–distortion–accuracy approaches, in principle, optimize quantization steps for JPEG and JPEG 2000 encoders to minimize the classification loss and bitrate.

With the remarkable success of learned image compression, there has been a surge of interest in visual compression for machine vision, aiming to preserve machine task performance on compressed data [23,143,144]. Le et al. [143] proposed the first end-to-end learned system that optimizes the rate–distortion trade-off, but the distortion includes the training loss of the pre-trained neural network task. Their proposed system architecture of image coding for machines is conceptually illustrated in Figure 18. The neural network-based codec outperforms the state-of-the-art VVC standard on object detection and instance segmentation tasks, achieving a BD rate gain of −37.87% and −32.90%, respectively. Le’s approach remains as performing the machine learning task in the reconstructed domain. Their results show that backgrounds are not prioritized with the bit budget because the instance segmentation task is not sensitive to those regions.

Le et al. [23] proposed an inference-time content-adaptive fine-tuning scheme that optimizes the latent representation to improve compression efficiency for machine consumption. Wang et al. [144] proposed an inverted bottleneck structure for the encoder and investigated the design and optimization of the network architecture that could be further improved for compression towards machine vision.

Most Compress-then-Analyze approaches are typically task-agnostic, meaning we cannot determine whether a reconstructed image $\hat{x}$ optimized for one task, such as classification, retains sufficient information for another task, such as segmentation. Dubois et al. [145] addressed this issue by characterizing the bitrate required to ensure high performance on all invariant predictive tasks under a set of transformations. They designed unsupervised objectives for training neural compressors and trained a generic image compressor that outperforms JPEG by orders of magnitude on eight datasets, even those it was never trained on (i.e., zero-shot).

### 4.2. Compressed Domain Analysis

Some works [58,59,60] attempted to learn from the compressed domain with comparative or better accuracy than the reconstructed domain. Singh et al. [59] proposed an end-to-end framework to learn compressible features, equivalent to learning from compressed latent representations. In a typical supervised classification problem, where *x* is the input image pixels, *y* is the target label, and $\hat{y}$ is the prediction, they minimized a loss function $L(y,\hat{y})$ given by:(22)$$L\left(y,\hat{y}\right)=L\left(y,f\left(x\right)\right),$$
where f(·) denotes the task neural network used to predict $\hat{y}$. For applications utilizing a pre-trained neural network, they removed the top few layers, and the remaining topmost layers were denoted as $f_{z}\left(\cdot \right)$, serving as the feature extractor. The output of the feature extractor is the latent representation $z=f_{z}\left(x\right)$, which replaces *x* in predicting $\hat{y}$:(23)$$\hat{y}=f_{\hat{y}}\left(z\right)=f_{\hat{y}}\left(f_{z}\left(x\right)\right).$$

Here, the task network f(·) is split into the feature extractor $f_{z}\left(\cdot \right)$ and the prediction network $f_{\hat{y}}\left(\cdot \right)$. Singh’s work aimed to learn a model $f_{\hat{y}}\left(f_{z}\left(\cdot \right)\right)$ such that *z* is compressible while still maintaining the performance of the original network f(·) by optimizing:(24)$$L(y,f_{\hat{y}}\left(f_{z}\left(x\right)\right))+\lambda R\left(z\right),$$
where *R* is the rate function as compression loss and λ is the trade-off parameter. The rate function *R* encourages the compressibility of *z* by penalizing an approximation of its entropy. From another perspective, the prediction network $f_{\hat{y}}\left(\cdot \right)$ learns from a compressed latent representation *z*.

Duan et al. [24] employed transfer learning to perform semantic inference directly from quantized latent features in the deep compressed domain without pixel reconstruction. In addition to the classification problem, various techniques such as image data hiding [21], image denoising [22], and image super-resolution [146] have been developed to operate directly on neural compressed latent spaces. Regarding end-to-end learned compression, neural network-based data processing and machine analytic tasks can operate in the compressed domain and be jointly optimized without decoding first. In some cases, the reconstruction task for human vision is optional.

### 4.3. Scalable Image Coding for Human and Machine

A scalable codec [147] is a type of codec that allows for the encoding and decoding of media data in a scalable manner. It can represent data at multiple quality levels or resolutions within a compressed bitstream, typically comprising one base layer and multiple enhancement layers. Singh et al. [59] have a similar concept of scalability by splitting the task network into a feature extractor and a prediction network as indicated in Equation (Equation 23). In Singh’s framework, the compressible features *z* are solely used for a machine task rather than for reconstruction.

Hu et al. [148,149] proposed a scalable feature representation for coding face images. In their approach, edge maps required for facial landmark detection form the base layer, while additional color information forms the enhancement layer. Facial landmark detection can be performed from the base layer alone, while facial images can be reconstructed using both the base and enhancement layers. Thus, a scalable learned image codec that caters to both machine and human vision is achieved.

Yan et al. [150] considered, from the perspective of information theory [151], that the deep features of an image naturally form an entropy-decreasing series: a scalable bitstream can be attained by compressing the features backward from a deeper layer to a shallower layer. They proposed the SSSIC framework to obtain an embedded bitstream that can be either partially decoded for semantic analysis or fully decoded for human vision.

Choi et al. [152] formulated latent space scalability to support object detection and input image reconstruction as a Markov chain model, illustrated in Figure 19. A neural encoder $G_{a}$ encodes the input image X to produce the compressed latent representation Y. At the receiver side, a neural decoder $G_{s}$ can reconstruct $\hat{X}$ from the latent code Y or use a task network F to perform a prediction as T. Additionally, the task network F can operate on the reconstructed image $\hat{X}$ to obtain T, considering that it is quite common for an object detector to operate on a JPEG-compressed image.

The processing chain $X\to Y\to \hat{X}$ represents the image encoding and decoding process, while the chain $Y\to \hat{X}\to T$ denotes the typical machine task prediction. By applying the data processing inequality [151] to the prediction processing chain, we obtain
(25)$$I\left(Y;\hat{X}\right)\geq I\left(Y;T\right),$$
where I(·;·) denotes the well-known mutual information [151]. This inequality suggests that intermediate features Y carry less information about task T than input reconstruction $\hat{X}$. Consequently, we construct features Y such that only a portion of Y is used for task prediction while the entire Y is used for input reconstruction to become feasible. Choi et al. [153] later extended this idea to propose a scalable image coding framework based on well-developed neural compressors, achieving up to 80% bitrate savings for machine vision tasks. Ozyilkan et al. [154] further improved Choi’s latent space scalability by learning two disentangled latent representations for a more compact latent representation than [153].

### 4.4. International Standards

Until now, the concept of Video Coding for Machines (VCM) has referred to learned compression methods targeting machine vision only or both machine and human vision. As described earlier, with the end-to-end learned approaches, the field of compression and visual analysis has evolved from pixel domain analysis to compression for analysis, analysis in the compressed domain, and compression for humans and machines. Efficient VCM has become an essential topic in academia and industry. Several papers review task-oriented video coding [155] and intelligent video coding [156].

In alignment with this trend, international standard committees have launched initiatives such as MPEG Video Coding for Machines (MPEG VCM) [26] and JPEG AI [27], aiming to bridge the fields of data compression and computer vision. This section will describe the two VCM standards to illustrate the industry’s efforts in this direction.

#### 4.4.1. MPEG VCM

In July 2019, the international standard organization MPEG established an ad hoc group named VCM to study the requirements for potential standardization work. The MPEG VCM activity [26] aims to standardize a bitstream format generated by compressing a previously extracted feature stream alongside an optional video stream. The bitstream should enable multiple machine vision tasks. VCM shall be able to perform the following:Efficiently compress the bitstream: the size of the compressed features shall be less than that of the encoded video stream using state-of-the-art video compression technologies like HEVC and VVC.Use the bitstream to support single or multiple tasks. Features should be general enough for different scenarios, such as object detection and segmentation.Support varying performance levels for multiple tasks as measured by the appropriate metrics. This performance level may depend on the application.Allow the reconstruction of the original input for human consumption. This may be achieved with an additional bitstream.

MPEG VCM has identified relevant use cases and related requirements [157], focusing on machine-to-machine communication in intelligent transportation and the hybrid machine and human consumption for surveillance and smart city use cases. Figure 20 illustrates a potential VCM architecture. The VCM codec could be a video codec, a feature codec, or both. In the case of a feature codec, VCM feature encoding encompasses feature extraction, feature conversion/packing, and feature coding. An interface to an external Neural Network Compression and Representation might exist for feature extraction and task-specific networks. The MPEG VCM group has identified six classes of use cases [157], including intelligent transportation, smart city, and intelligent content. Moreover, the MPEG VCM group has established an evaluation framework that includes computer vision tasks [158]. Typical tasks include object detection, object segmentation, object tracking, action recognition, and pose estimation. Since its inception, the VCM group has been presented with various technical solutions, and responses to the Call for Evidence (CfE) have been detailed in Guo’s recent update [159] on VCM standard development.

MPEG VCM is presently undergoing standardization in two distinct areas: feature coding (Track 1) and image and video coding (Track 2). Both tracks demonstrated superior performance compared to a VVC anchor. Therefore, we believe that the standardization of VCM will proceed smoothly according to the planned schedule. MPEG VCM is expected to replace conventional video coding technologies in various fields where machine vision is a must [160].

#### 4.4.2. JPEG AI

In March 2019, the Joint Photographic Experts Group (JPEG) committee requested a survey of the state-of-the-art learning-based image coding methods [161] as the initial step in the standardization process within JPEG. Subjective quality evaluations of early solutions [162] show that learning-based image codecs produce very different artifacts as compared with conventional codecs but generally remain competitive in subjective quality. Subsequently, the JPEG AI standard was initiated to develop the first image coding standard based on machine learning [163]. The scope of the JPEG AI standardization is as follows:Create a learning-based image coding standard.Offer a single-stream, compact compressed domain representation, targeting human visualization, with significant compression efficiency over conventional image coding standards at equivalent subjective quality.Provide effective performance for image processing and computer vision tasks.Support a royalty-free baseline.

Figure 21 illustrates the high-level JPEG AI framework cited from the JPEG AI white paper [27], featuring three pipelines. The input to the learning-based image coding framework is a digital image, and the output bitstream may undergo standard reconstruction for human visualization, producing a standard decoded image. Alternatively, in the other pipelines, standard reconstruction may be omitted, as the latent representation contains the necessary information not only for decoding but also for executing image processing and computer vision tasks on the decoder side. These tasks are performed on the latent representation rather than the lossy decoded image. This inherently feature-rich latent representation can be used in two primary ways: (1) for executing an image processing task, such as enhancing or modifying the image, resulting in a processed image, and (2) for executing a computer vision task, extracting high-level semantic information.

JPEG AI has identified three key tasks besides standard reconstruction: (1) compressed domain image classification, (2) compressed domain super-resolution, and (3) compressed domain denoising. More compressed domain decoders are expected to be developed in the future. To fulfill the scope outlined by JPEG AI, the corresponding encoder must generate a bitstream independent of the task, ensuring that the latent representation enables efficient processing by a compressed domain decoder. In this context, the term “decoder” refers to not only the process of translating the bitstream to pixel data but also other modalities, such as converting images to textual labels. The desirable requirements for JPEG AI are elaborated in [164].

Presently, JPEG AI’s focus remains on the standard reconstruction task. Regarding coding efficiency, JPEG AI reference software VM1 can achieve average BD-rate gains over VVC Intra of 28% for the tools-off configuration. Additionally, the subjective quality improvement of JPEG AI VM over VVC/H.266 Intra is evident. Regarding complexity, JPEG AI VM1 can achieve a 30% compression gain over VVC Intra operating at 200 kMAC/pixel (which may become acceptable for many devices in the mid-term) and a 15% compression gain operating at just 20 kMAC/pixel.

As per the timeline outlined in [165], the JPEG AI standard [166] has been divided into two versions. Version 1 addresses several (but not all) JPEG AI core and desirable requirements with an emphasis on standard reconstruction as defined in [164]. Version 2 encompasses multiple objectives, including JPEG AI requirements yet to be addressed in Version 1 and improved solutions for existing JPEG AI requirements. The Version 1 JPEG AI Core Coding System is expected to become the first international standard by October 2024.

## 5. Conclusions

In this paper, we conduct a comprehensive survey and review of lossy image compression methods, including both conventional codecs and modern end-to-end learned image compression approaches. Our focus lies mainly on lossy image compression due to its imperative role in achieving superior compression rates, which are essential for machine vision tasks in the AI and smart city era. We dedicate attention to learning-based image coding techniques pertinent to meeting the coding requirements for human- and machine-centric applications.

The remarkable success of learned image compression methods offers great flexibility and adaptability across various new data domains while supporting different distortion losses. Moreover, the internal representations of learned image compression methods naturally lend themselves to learning-based data processing and analytic pipelines. Compression and visual analysis fields have witnessed a transformative evolution, transitioning from pixel domain analysis to compression for analysis, analysis in the compressed domain, and compression for both humans and machines. Consequently, the concept of Video Coding for Machines has emerged from both academic research and industry applications.

We also provide a concise overview of two prominent international standards, MPEG Video Coding for Machines and JPEG AI, which aim to bridge the gap between data compression and computer vision for practical industry use cases. Despite the challenges faced by learned image compression methods, such as computational complexity, ongoing research endeavors will swiftly address these issues, thereby paving the way for a unified future visual coding paradigm catering to both human and machine needs.

## Figures and Tables

**Figure 1 entropy-26-00357-f001:**
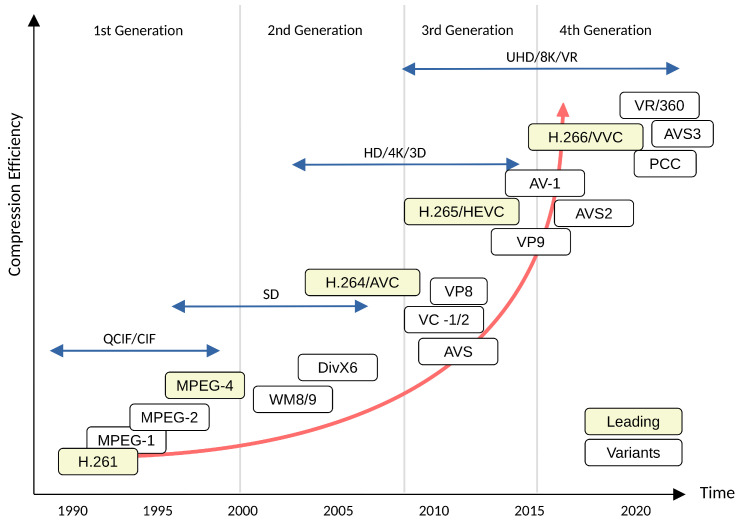
Evolution of video coding standards since the 1990s.

**Figure 2 entropy-26-00357-f002:**
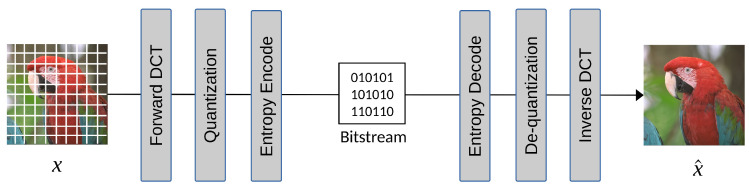
The high-level JPEG encoding and decoding flows.

**Figure 3 entropy-26-00357-f003:**
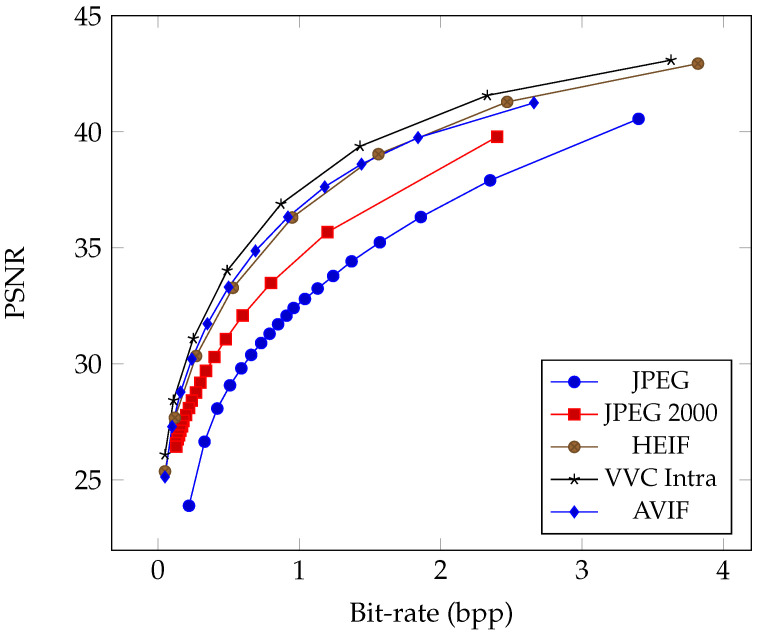
The rate–distortion curve comparison of modern image codecs since JPEG. Evaluated on the Kodak dataset.

**Figure 4 entropy-26-00357-f004:**
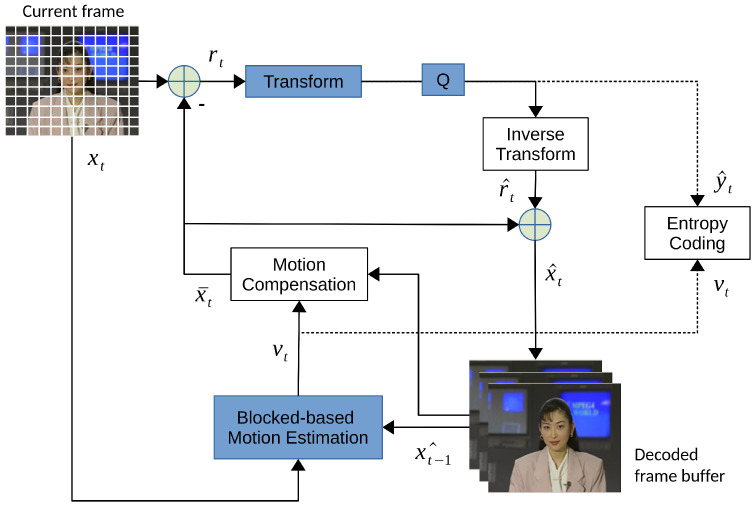
A simplified traditional video coding flow.

**Figure 5 entropy-26-00357-f005:**
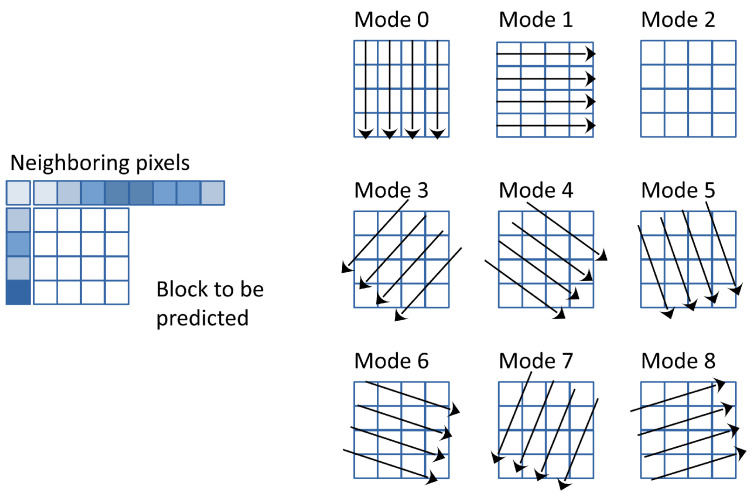
The nine intra 4×4 prediction modes are defined in the H.264 standard.

**Figure 6 entropy-26-00357-f006:**
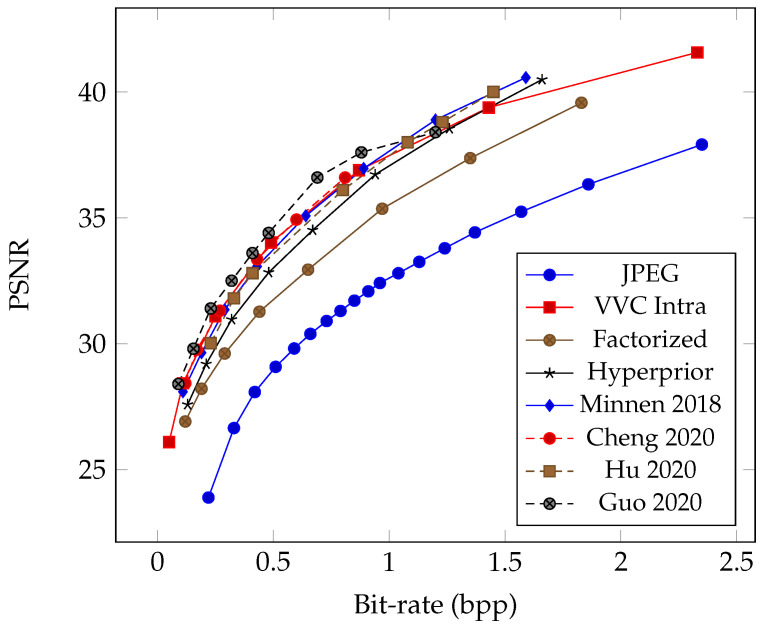
The rate–distortion curves of various learned compression methods compared with JPEG and VVC Intra. Evaluated on the Kodak dataset [12,13,14,17,18,20].

**Figure 7 entropy-26-00357-f007:**
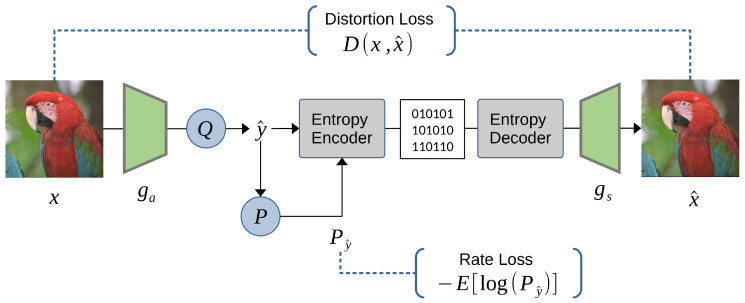
A typical autoencoder-based end-to-end learned image compression flow.

**Figure 8 entropy-26-00357-f008:**
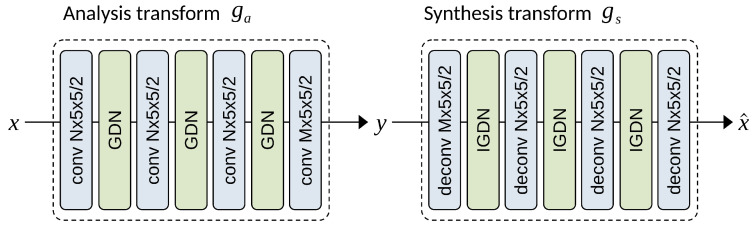
The network architecture of the analysis and synthesis transform mimics the traditional image transform proposed in the hyperprior model [13].

**Figure 9 entropy-26-00357-f009:**
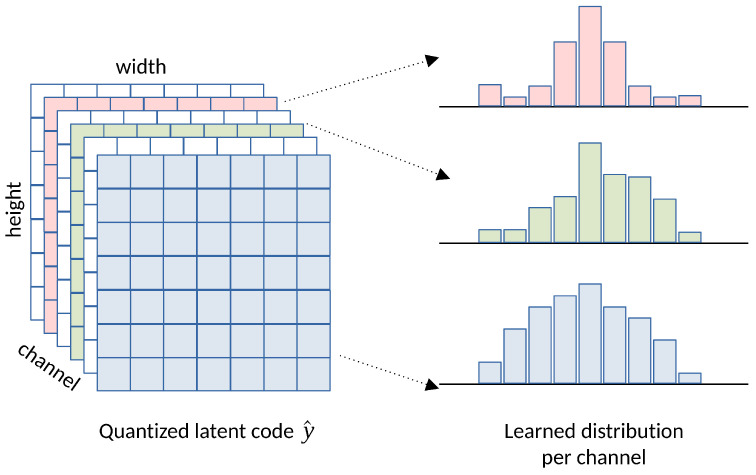
The factorized entropy model.

**Figure 10 entropy-26-00357-f010:**
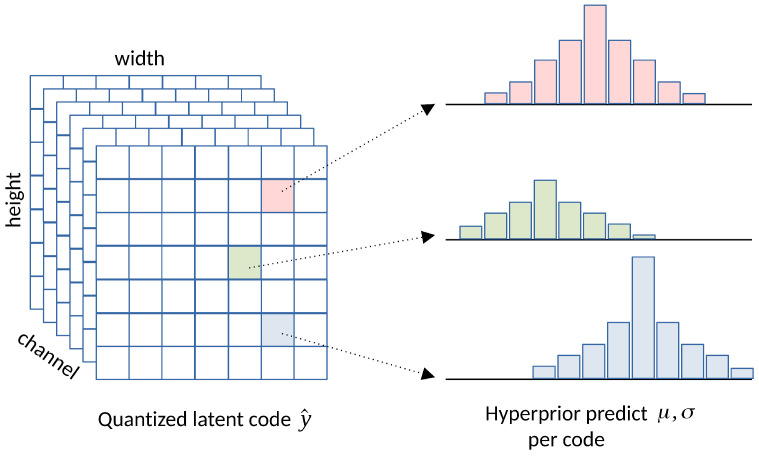
The hyperprior entropy model.

**Figure 11 entropy-26-00357-f011:**
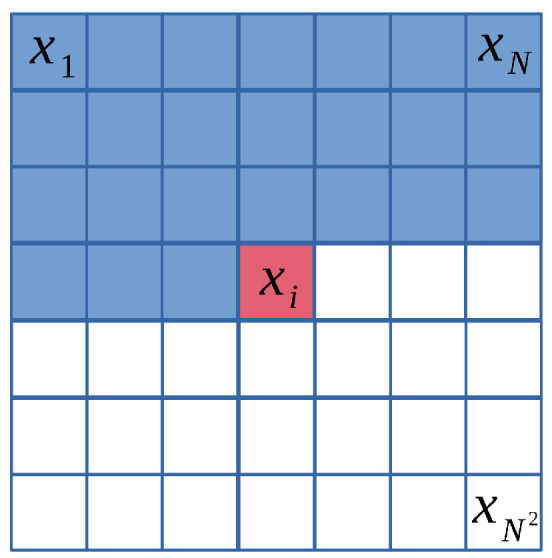
The pixel-based context model.

**Figure 12 entropy-26-00357-f012:**
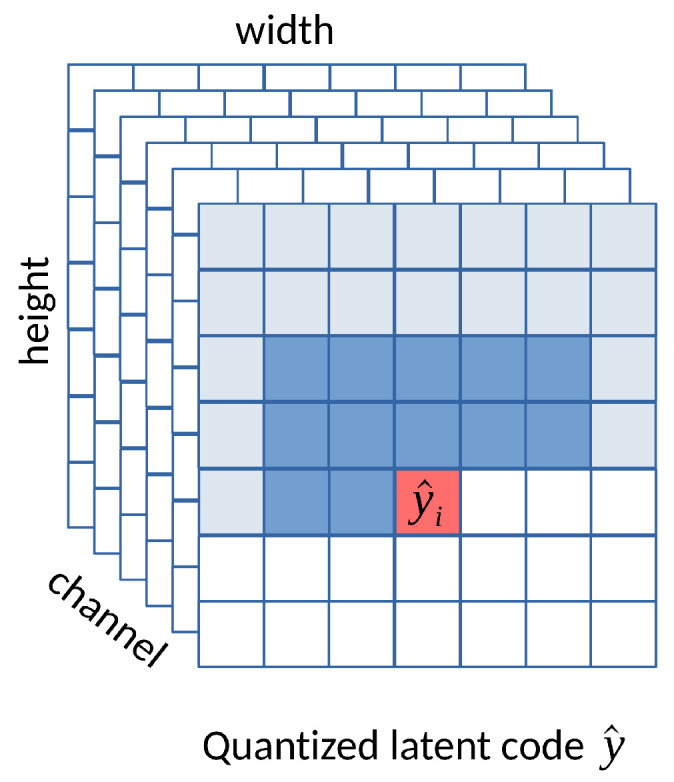
The joint autoregressive context model and hyperprior model.

**Figure 13 entropy-26-00357-f013:**
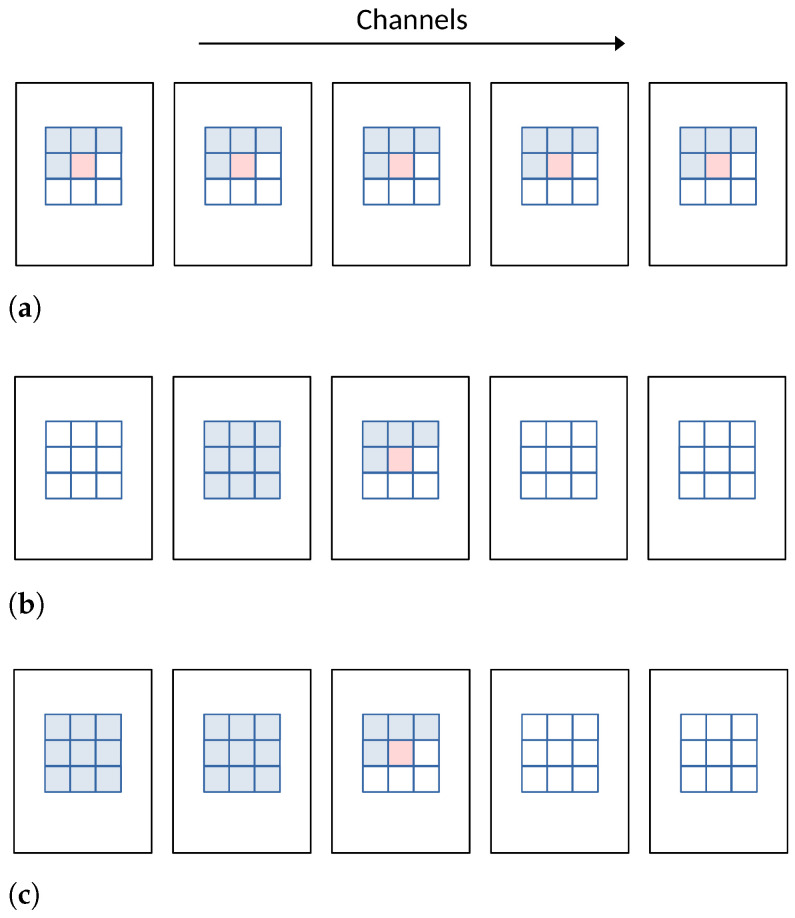
Different context models for latents in deep image compression, where the red blocks represent current latent elements, and the blue blocks represent the latent elements used as contexts for current latent elements. (**a**) Minnen et al. [14]: the 2D spatial context model. (**b**) Mentzer et al. [72]: the 3D masked convolution context model. (**c**) Ma et al. [73]: the cross-channel context model.

**Figure 14 entropy-26-00357-f014:**
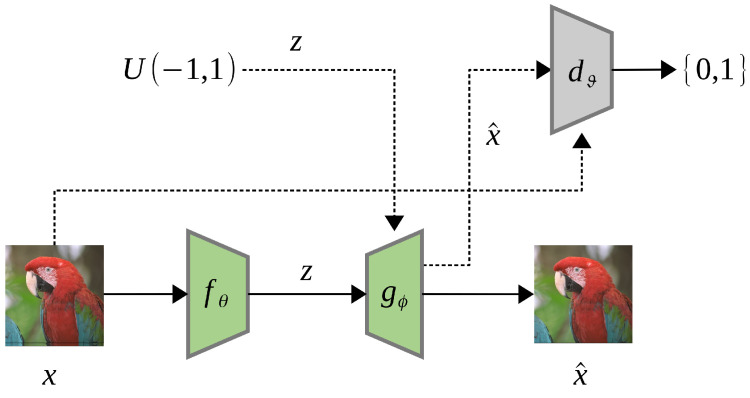
The generative compression architecture proposed by Santurkar et al. [88]. The dot lines represent the adversarial training.

**Figure 15 entropy-26-00357-f015:**
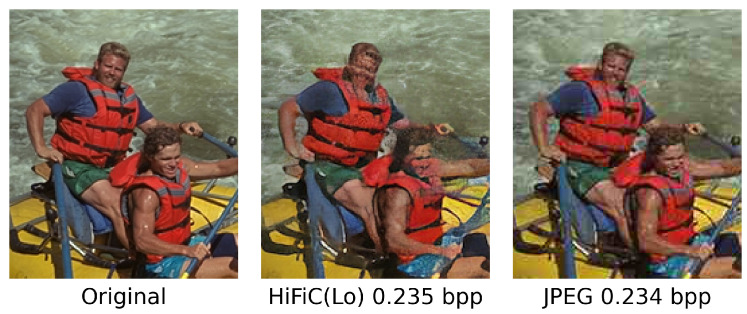
The reconstructed result of HiFiC [93] at a low bitrate compared to the JPEG. Note that the faces are seriously distorted.

**Figure 16 entropy-26-00357-f016:**
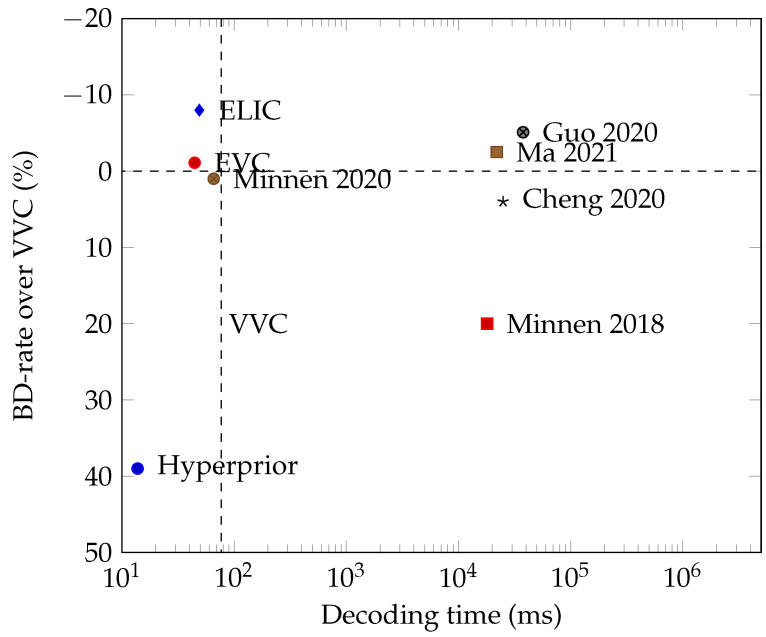
The “Bitrate-speed” comparison of various learned image compression methods on the Kodak dataset (The better results are approaching the left-top corner) [13,14,17,20,73,74,127,128].

**Figure 17 entropy-26-00357-f017:**
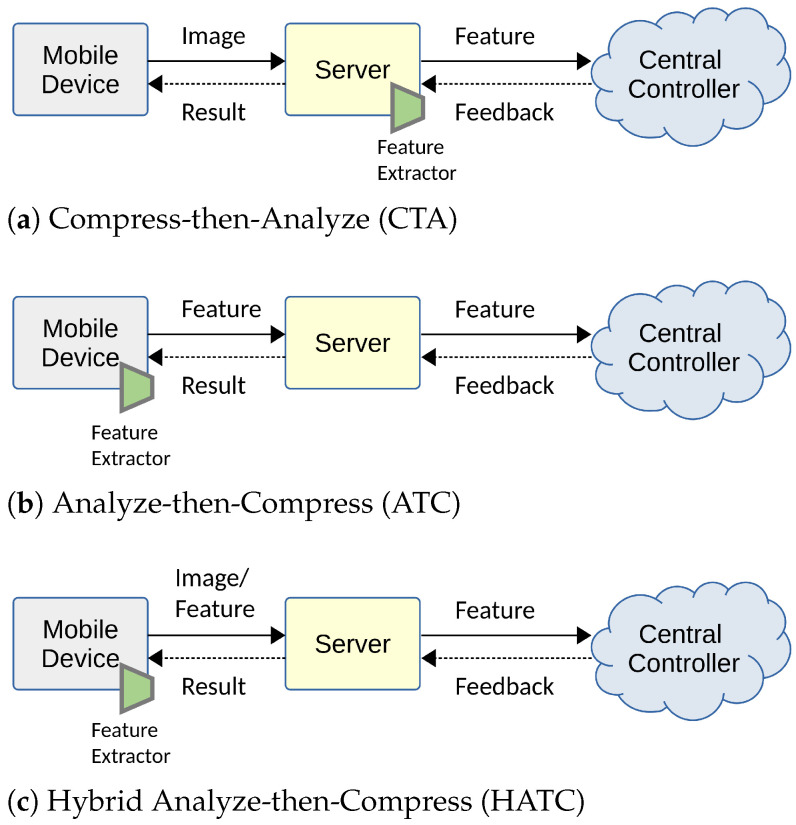
Architectures of visual retrieval applications regarding analysis and compression.

**Figure 18 entropy-26-00357-f018:**
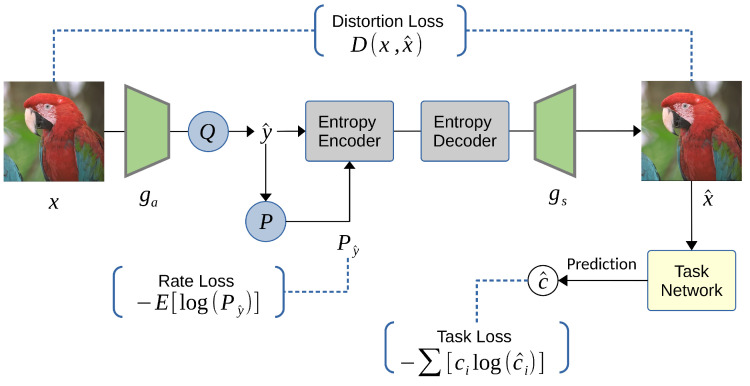
The architecture of image coding for machines that was proposed in [143].

**Figure 19 entropy-26-00357-f019:**
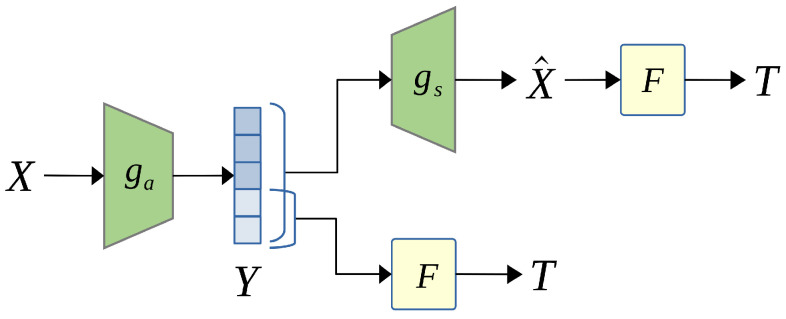
Markov chain model for a scalable learned compression system.

**Figure 20 entropy-26-00357-f020:**
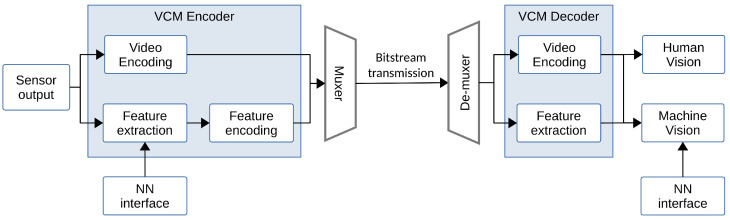
The high-level illustration of the potential VCM architecture. In this figure, the NN interface represents the input from external neural network systems.

**Figure 21 entropy-26-00357-f021:**
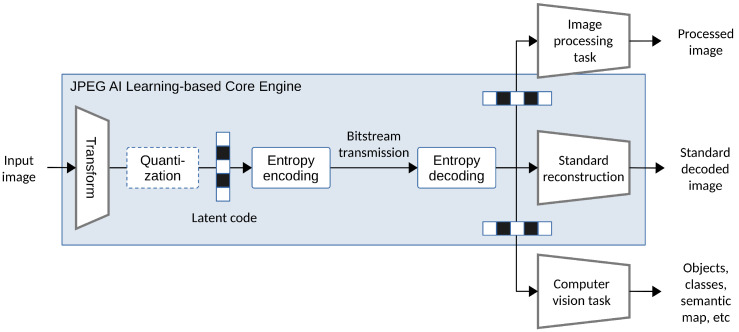
JPEG AI learning-based image coding framework.

**Table 1 entropy-26-00357-t001:** Summary of prior learned image and video compression review papers.

Survey Paper	Image	Video	Focus	Notes
[28] Ma et al. (2019)	✓	✓	E2E-ML, Hybrid	Review neural network-based methods in the early days.
[34] Zhang et al. (2020)		✓	Hybrid	Focus on ML-assisted video coding. Review ML-based visual quality assessment methods.
[33] Liu et al. (2020)	✓	✓	E2E-ML, Hybrid	Propose a case study of a hybrid video codec.
[35] Hoang et al. (2021)		✓	E2E-ML, Hybrid	
[29] Hu et al. (2021)	✓		E2E-ML	Provide a detailed learned image codec benchmark.
[36] Yang et al. (2022)	✓	✓	E2E-ML	Cover lossless image compression and information theory foundations.
[30] Mishra et al. (2022)	✓		E2E-ML, Hybrid	Categorize prior arts by network architectures.
[31] Jamil et al. (2023)	✓		E2E-ML	
[32] Chen et al. (2024)	✓		E2E-ML	Review visual compression with deep generative models in ultra-low bitrate communication.

**Table 2 entropy-26-00357-t002:** Summary learned image compression methods by categories and notes for pros and cons.

Category	Method	Pros	Cons	Notes
Pixel probability models	[66,67,68,69,70,71]	Simple and intuitive modeling	Sequential process is slow and cannot be parallelized	Predictive coding
AE-based transforms	[12,13]	Efficient architecture for satisfying coding efficiency	Remain inferior to modern codec such as HEVC	Transform coding
Context models	[14,15,17,20,72,73,74] ^1^	Coding efficiency surpasses HEVC and VVC	Model complexity is too high and slow	Predictive and transform coding
Multi-scale approaches	[16,18,75,76] ^2^	Competitive coding efficiency with less computational complexity		Wavelet-liked transform coding
Attention and Content-based approaches	[17,19,72,77]	Straightforward to enhance coding performance	Gains may not significant if underlying entropy model is complex	
Transformer based	[80,81,82,83,84,85]	Better compression efficiency than CNN-based transforms, requiring fewer parameters and shorter decoding time		
Generative compression	[75,88,91,92,93,96,97,98]	Generates visually pleasing images with very low bitrate	Semantic regions may be seriously distorted	

^1^ First learned image codec that surpasses VVC. ^2^ Achieves state-of-the-art compression performance with low computational complexity.

## Data Availability

No new data were created or analyzed in this study. Data sharing is not applicable to this article.

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
