# Peer review of "Unveiling the Future of Human and Machine Coding: A Survey of End-to-End Learned Image Compression"

_entropy, 2024, doi:10.3390/e26050357_

Round 1
Reviewer 1 Report
Comments and Suggestions for Authors
(1) In the Introduction, the authors should summarize the differences between this review and the existing reviews, and summarize the contributions of this paper.
(2) The article not only describes the coding of an image but also covers video, but the title only focuses on the image, and whether the video coding should be added.
(3)This paper mainly describes learned compression. Is it necessary to introduce traditional coding techniques?
(4)A list should be added to summarize the pros and cons of the various learned compression methods and where they can be used.
Author Response
Thanks for the anonymous reviewer's valuable feedback. Our replies are presented below regarding the comments in sequence:
1. It’s true that we could better summarize our contribution at the beginning. We’ve revised the last paragraph of Section 1.1.
2. We think discussing conventional video codecs in Section 2 remains necessary because recent image coding standards are mostly built on video intra-frame coding techniques. The concept of more accurate intra-frame predictive coding contributes the most to the post-JPEG codecs. Moreover, since the development of learned image codecs starts with migrating a JPEG-like flow and then advances the techniques with more accurate intra-prediction, the brief introduction to modern video codecs could give readers a more holistic view. As a result, we revised Section 1.2 to reflect our thoughts.
3. Same as point 2. The development of learned image codecs can be thought of as replacing hand-crafted coding tools one by one with learned approaches. So, it’s necessary to describe conventional codecs to be self-completed.
4. We agree, and points are taken. We revised the first paragraph of Section 3.8 to indicate and added Table 2 to summarize learned image compression methods by categories and notes for pros and cons.
The revised manuscript is attached to this reply. We mark the changes in red so that it's easy to spot and check.

Reviewer 2 Report
Comments and Suggestions for Authors
This is a well written review paper on the subject of End-to-end Learned Image Compression methods. I have the following suggestions to improve the manuscript:
1. Since the review is only on lossy compression, I would suggest that in Section 1 of the paper the authors give a short write-up on the different types of compression - lossy vs. lossless and a bit of write-up on Rate-Distortion curve. A review paper should set the context appropriately before plunging into the details of learned image compression methods.
2. Is it possible to give a timeline graph of breakthroughs, milestones in lossy and/or lossless compression methods since Shannon 1948? I think a visual representation of all the key methods/papers with years and author names would be appealing. I am making this suggestion since this is mainly a review paper and not an original article. This will improve readability greatly.
3. I am not very satisfied with the subsection on Computational Complexity. It would be good to give a table comparing the order of complexities of various methods (at least the important ones). Now, the section is lost in the verbose details. A table would greatly help.
4. Similarly, for the image quality aspect of lossy compression - a table on what are the different metrics that have been developed so far to evaluate image quality of lossy compressors can be given.
5. A timeline/table of different image/video compression standards - when were they setup/released with dates and acronyms can be given to make it easy for the reader
Since this is a survey paper, readability and easy access to information such as milestones, key papers, important years, breakthroughs in performance should be visually presented. This makes a survey/review paper very appealing and useful to the field.
Author Response
Thanks for the anonymous reviewer's valuable feedback. Our replies are presented below regarding the comments in sequence:
1. We agree, and points are taken. We've revised the second paragraph of Section 1.
2. In our humble opinion, the field of data compression is so broad that reviewing traditional lossless and lossy methods can be at least 10 pages for each. We choose to focus on lossy compression in this survey paper, as we explained in the last paragraph in Section 1.1. We are unsure if we put breakthroughs and milestones of entropy coding, predictive coding, etc., and all the end-to-end learned techniques in one visual representation would help readability. However, the reviewer's suggestion remains valid. We add Figure 1 to summarize the progress of traditional video codecs (they are relevant to image compression because HEIF and AVIF come from video intra-coding). Figure 6 is added to summarize the progress of learned image compression milestones. We revised the first paragraph of Section 3.8 and added Table 2 to summarize learned image compression methods by categories and notes for pros and cons. We hope that could help with the overall readability.
3. We agree with the complexity part. We revised Section 3.8.1 and added Figure 16 to illustrate the bitrate-speed comparison of learned image compression methods.
4. Regarding the quality metrics for lossy image compression, the PSNR and MS-SSIM are still the most popular as traditional metrics. Learning-based metrics such as LPIPS and FID often measure generative compression methods. These quality metrics are not purposely developed to measure lossy compression. In other words, when we develop a lossy compression, we choose a quality metric from known ones so that it can adequately measure the distortion characteristics caused by our codec. If a golden metric can highly correspond to human perception, we will use that metric and optimize for that. Unfortunately, it is unknown, as we indicated in section 3.8.2.
5. We agree and points taken. We add Figure 1 to summarize the traditional video codecs.
The revised manuscript is attached to this reply. We mark the changes in red so that it's easy to spot and check.

Reviewer 3 Report
Comments and Suggestions for Authors
The authors present an overview of the state-of-the-art image/video coding techniques that use
machine learning (ML) and neural networks. Although this is not the first overview of the topic,
the paper looks worth publishing since it covers all aspects of the problem and the quality
of presentation is rather high.
Remarks
1. Numerical results for ML-based techniques are not summarised in the form of tables or figures.
A plot similar to that inFig. 2 would help to understand the progress due to new technologies.
2. The complexity issues are not touched. An additional paragraph explaining
perspectives of replacing classical techniques with new ones in real-life applications would be suitable.
Author Response
Thanks for the anonymous reviewer's valuable feedback. Our replies are presented below regarding the comments in sequence:
1. We agree with this valuable suggestion. Figure 6 is added to summarize the progress of learned image compression milestones.
2. We agree with the complexity part. We revised Section 3.8.1 and added Figure 16 to illustrate the bitrate-speed comparison of learned image compression methods.
The revised manuscript is attached to this reply. We mark the changes in red so that it's easy to spot and check.

Round 2
Reviewer 1 Report
Comments and Suggestions for Authors
The authors have been addressed my questions, so i suggest to accept it for publish.
Reviewer 3 Report
Comments and Suggestions for Authors
I recommend acceptance